# PIP$_2$ regulating calcium signal modulates actin cytoskeleton-dependent cytoadherence and cytolytic capacity in the protozoan parasite *Trichomonas vaginalis*

**Yen-Ju Chen**[1] [☯], **Kuan-Yi Wu**[1] [☯], **Shu-Fan Lin**[1] [☯], **Sung-Hsi Huang**[1,2], **Heng-Cheng Hsu**[3,4,5], **Hong-Ming Hsu**[1] *

**1** Department of Tropical Medicine and Parasitology, National Taiwan University College of Medicine, Taipei, Taiwan, **2** Department of Internal Medicine, National Taiwan University Hospital Hsin-Chu Branch, Hsinchu, Taiwan, **3** Department of Obstetrics and Gynecology, National Taiwan University Hospital and National Taiwan University College of Medicine, Taipei, Taiwan, **4** Department of Surgery, National Taiwan University Cancer Center, Taipei, Taiwan, **5** Graduate Institute of Clinical Medicine, National Taiwan University College of Medicine, Taipei, Taiwan

☯ These authors contributed equally to this work.

* hsuhm@ntu.edu.tw

**Data Availability Statement:** All relevant data are within the manuscript and its Supporting Information files.

## Abstract

*Trichomonas vaginalis* is a prevalent causative agent that causes trichomoniasis leading to uropathogenic inflammation in the host. The crucial role of the actin cytoskeleton in *T. vaginalis* cytoadherence has been established but the associated signaling has not been fully elucidated. The present study revealed that the *T. vaginalis* second messenger PIP$_2$ is located in the recurrent flagellum of the less adherent isolate and is more abundant around the cell membrane of the adherent isolates. The *T. vaginalis* phosphatidylinositol-4-phosphate 5-kinase (*Tv*PI4P5K) with conserved activity phosphorylating PI(4)P to PI(4, 5)P$_2$ was highly expressed in the adherent isolate and partially colocalized with PIP$_2$ on the plasma membrane but with discrete punctate signals in the cytoplasm. Plasma membrane PIP$_2$ degradation by phospholipase C (PLC)-dependent pathway concomitant with increasing intracellular calcium during flagellate-amoeboid morphogenesis. This could be inhibited by Edelfosine or BAPTA simultaneously repressing parasite actin assembly, morphogenesis, and cytoadherence with inhibitory effects similar to the iron-depleted parasite, supporting the significance of PIP$_2$ and iron in *T. vaginalis* colonization. Intriguingly, iron is required for the optimal expression and cell membrane trafficking of *Tv*PI4P5K for in situ PIP$_2$ production, which was diminished in the iron-depleted parasites. *Tv*PI4P5K-mediated PIP$_2$ signaling may coordinate with iron to modulate *T. vaginalis* contact-dependent cytolysis to influence host cell viability. These observations provide novel insights into *T. vaginalis* cytopathogenesis during the host-parasite interaction.

**Funding:** This study was supported by grants from the National Science and Technology Council (110-2320-B-002-076- and 112-2320-B-002-060-, Grant Recipient: Dr. Hong-Ming Hsu) and National Taiwan University (NTU-JP-112L7226, Grant Recipient: Dr. Hong-Ming Hsu). The funders had no role in study design, data collection and analysis, decision to publish, or preparation of the manuscript.

**Competing interests:** The authors have declared that no competing interests exist.

## Author summary

Signal transduction in *T. vaginalis* pathogenesis remains elusive despite elucidation of the distinctive actin-based cytoadherence of the adherent isolates. Herein, we identified a *T. vaginalis* phosphatidylinositol-4-phosphate 5-kinase (*Tv*PI4P5K) which converts PI(4)P into PI(4, 5)P$_2$. Differential *Tv*PI4P5K expression and specific PIP$_2$ localization in the *T. vaginalis* isolates with distinct cytoadherence suggest a potential link underlying PIP$_2$ signaling and cytoskeleton behaviors in this parasite. In the parasite transforming from the flagellate into an amoeboid trophozoite, the plasma membrane PIP$_2$ signal may be cleaved by a phospholipase C (PLC)-like pathway, thereafter increasing intracellular calcium, both of which are required for parasite actin assembly, morphogenesis, and cytoadherence. Moreover, environmental iron elicited *Tv*PI4P5K and PIP$_2$ expression and triggered *Tv*PI4P5K translocation to the plasma membrane. PIP$_2$ signaling also modulated the extracellular cytopathic effectors from the parasite to lyse host cells. Our study reveals the biological significance of PIP$_2$ signal transduction in regulating *T. vaginalis* pathogenesis, which may be a potential therapeutic target.

## Introduction

*Trichomonas vaginalis* is a pathogenic protozoan that causes Trichomoniasis, a non-viral sexually transmitted disease prevalent worldwide. This extracellular parasite colonizes the host urogenital tract by adhering to the mucosa, causing uropathogenic inflammation. *T. vaginalis* interacts with host epithelium cells through numerous surface adhesion molecules [1,2,3,4,5,6], and its surface saccharide moiety is involved in the hemolysis of erythrocytes [7] and phagocytosis of the host cells [8,9]. Trichomonads lyse host cells and phagocytose cell debris for nutrient acquisition, concomitantly damaging the mucosal layer resulting in various cytopathic effects [10,11], including uropathogenic inflammation with worsening symptoms during menstruation. Multiple virulence factors are secreted from or bind to the *T. vaginalis* cell membrane to modulate parasitic cytoadherence, cytotoxicity, immunoglobulin degradation, and host cell apoptosis [5,12,13,14]. A recent study proposed that *T. vaginalis* may export extracellular cysteine peptidase through an unconventional lysosomal pathway [15].

Iron is an element essential for growth but an excess is toxic. An abundant element during menstruation, iron multifacetedly regulates *T. vaginalis* histone modification [16], transcription factor nuclear import [17], morphology [18], the expression of various genes involved in metabolism [16,19], cytoadherence [16,20], and proteolysis [21,22]. The *T. vaginalis* resistance to complement lysis is attributed to the expression or secretion of multifarious surface proteases with diverse effects on pathogenesis, which are also associated with the environmental iron concentration [21,22,23]. In *T. vaginalis* signal transduction, iron transiently activates PKA signaling to phosphorylate the Myb3 transcription factor and trigger ubiquitination required for its nuclear translocation and regulation of adhesin expression [17].

The *T. vaginalis* adherent isolate exhibits remarkable flagellate-amoeboid transition and cytoadherence, which is repressed by actin polymerization inhibitors, to slow down amoeboid migration and reduce cytoadherence, implying the critical roles of the actin cytoskeleton for host colonization. The actin cytoskeleton modulates cell behaviors such as morphological transition, protein trafficking and secretion, membrane adhesion, cell migration, and phagocytosis via diverse pathways between cell types [24,25,26]. Recent studies have demonstrated that the actin-cytoskeleton-mediated amoeboid morphogenesis capacity biologically correlates with cytoadherence and migration crucial for *T. vaginalis* colonizing host [27,28].

Phosphatidylinositol 4, 5 bisphosphates ($PIP_2$) is a versatile second messenger involved in cytoskeleton organization, vesicle trafficking, and transcription vital to polarized cell growth [29]. Intracellular $PIP_2$ is synthesized by either phosphatidylinositol 4-phosphate 5-kinase (PI4P5K) or phosphatidylinositol 5-phosphate 4-kinase (PI5P4K) using the substrates phosphatidylinositol-4-phosphate [PI(4)P] or phosphatidylinositol-4-phosphate [PI(5)P] respectively [30]. $PIP_2$ on the inner leaflet of the plasma membrane modulates actin cytoskeleton reorganization, including activating assembly accessory protein or branching factors, suppressing disassembly regulator activity, and accelerating the addition of G-actin at the barbed end of growing filaments, to fine-tune cytoskeleton dynamics [31]. By contrast, $PIP_2$ hydrolysis by phospholipase C (PLC) generates inositol 1,4,5-triphosphate ($IP_3$) and diacylglycerol (DAG) as the depolymerization signals. Subsequently, $IP_3$ provokes calcium-dependent signaling to activate severing proteins, such as gelsolins and cofilin, leading to actin meshwork disassembly [32]. The regulation of $PIP_2$ signaling cascades in the peripheral cytoskeleton remodeling differs in various cell types. Nonetheless, PI4P5K on the plasma membrane is essential for yeast cell morphogenesis [33], prompting us to investigate the link between the cytoskeleton and $PIP_2$ signaling in *T. vaginalis*.

This study identified differential $PIP_2$ and *Tv*PI4P5K expression in adherent and less adherent *T. vaginalis* isolates and analyzed their functional roles in *T. vaginalis* cytopathogenicity.

## Results

The crosstalk between the complicated regulatory complexes and access signaling around the plasma membrane coordinates cell morphogenesis. Recently, we found that actin cytoskeleton activity is involved in the morphogenesis and cytoadherence of *T. vaginalis* [27,28], prompting us to investigate the upstream signaling. T1 is a less adherent isolate with only the flagellate form freely swimming by flagellar motoring in suspension. TH17 is an adherent isolate with active flagellate-amoeboid transformation, tightly adhering to the solid surface of culture tubes with locomotion by amoeboid migration [27,28]. The two isolates with different cytoskeleton activities may help link $PIP_2$ signaling with cytoskeleton behaviors in *T. vaginalis*.

### Differential $PIP_2$ expression and localization in *T. vaginalis* isolates

Phosphatidylinositol (3,4,5)-trisphosphate ($PIP_3$) was undetectable in *T. vaginalis*, but the $PIP_2$ signal was observed in the cell membrane periphery of the TH17 flagellate trophozoites and the recurrent flagellum of the T1 less adherent isolate (Fig 1A). $PIP_2$ in the cell membrane or recurrent flagellum was also observed in NTU252 and NTU258 adherent isolates or G3 and NTU285 less adherent isolates, respectively (Figs 1B and S1). Overall, $PIP_2$ was more abundant in the adherent isolates than in the less adherent isolates (Fig 1C). In contrast to the long-term cultured experimental T1, G3, and TH17 isolates, the undomesticated short-term cultured NTU252, NTU258, and NTU285 clinical isolates freshly obtained from symptomatic vaginitis patients may preserve the innate parasite nature, suggesting that $PIP_2$ expression and distribution may be associated with parasite cytoadherence or locomotion. According to BLAST in TrichDB (https://trichdb.org/trichdb/app), only *Tv*PI4P5K (TVAG_462290) and *Tv*PI4P5K-2 (TVAG_456620) shared less than 30% sequence identity to human PI4P5K but with a conserved PI4P5K kinase domain (1~ 394 amino acid) and the consensus sequences for binding cell membrane, ATP, PI(4)P, $Mg^{2+}$, or $Mn^{2+}$ (S2 Fig) [34,35]. Quantitative PCR (qPCR) showed that *Tv*PI4P5K mRNA was highly expressed in the adherent isolates and positively correlated with $PIP_2$ abundance, whereas *Tv*PI4P5K-2 mRNA expression was equivalent in various isolates (Fig 1D), supporting the *Tv*PI4P5K function in $PIP_2$ production.

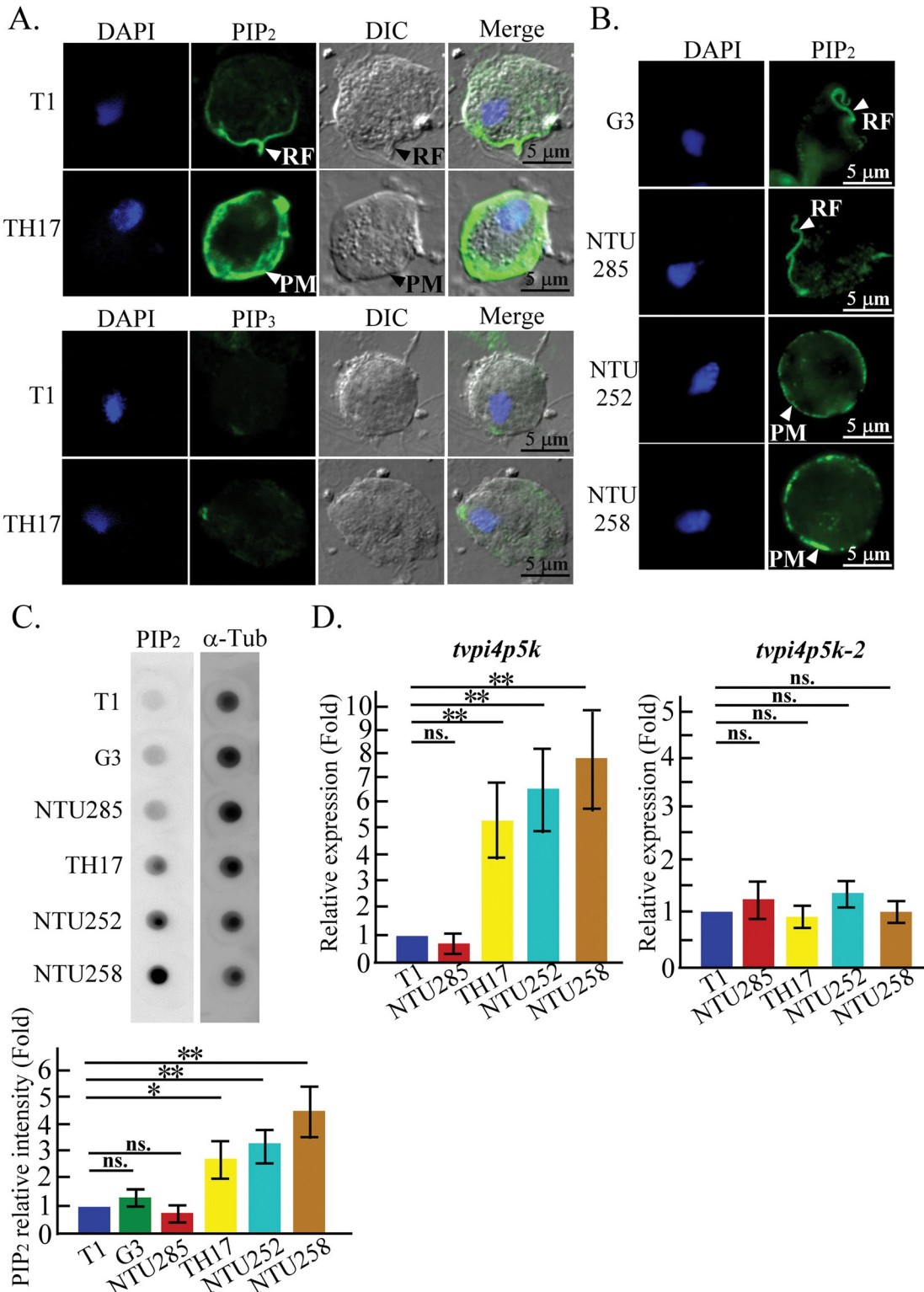

**Fig 1. Differential PIP$_2$ expression and localization in *T. vaginalis* isolates.** (A) The flagellate trophozoites of T1 and TH17 isolates were fixed for IFA detection by anti-PIP$_2$ (upper panel) or PIP$_3$ (lower panel) antibodies. (B) The G3 strain and NTU285, NTU252, and NTU258 clinical isolates were fixed and detected by anti-PIP$_2$ antibody detection for IFA. The arrowheads indicate recurrent flagellum (RF) or plasma membrane (PM). (C) The total lysates were subjected to a dot blot assay for PIP$_2$ and α-tubulin detection. The assay was performed in three biological repeats, and the relative intensity of PIP$_2$ normalized to α-tubulin signal is

shown in the bar graph (n = 3, mean ± SD). (D) qPCR was performed to quantify *tvpi4p5k* and *tvpi4p5k-2* gene transcription in various *T. vaginalis* isolates. The assay was performed in three biological repeats, and the relative gene expression normalized to the *β-tubulin* signals is shown in the bar graph (n = 3, mean ± SD). For (C) and (D), significant differences were analyzed by Student's t-tests, with $p < 0.05$(*) and $p < 0.01$ (**).

### *Tv*PI4P5K enzyme activity

To confirm the kinase activity of *Tv*PI4P5K, recombinant His-*Tv*PI4P5K with a wild-type kinase domain (1–414 aa) and a kinase-deficient K136A mutant were purified for the *in vitro* kinase assay (Fig 2A), revealing that Wt dose-dependently consumed ATP and was less affected by the K136A or BSA (Fig 2B), demonstrating that *Tv*PI4P5K uses ATP to phosphorylate PI(4)P to PI(4,5)P$_2$. Western blotting revealed that endogenous *Tv*PI4P5K expression in the adherent TH17 isolate was higher than in the less adherent T1 isolate (Fig 2C and 2D), in line with the qPCR results (Fig 1D), supporting the potential correlation of *Tv*PI4P5K and PIP$_2$ expression with *T. vaginalis* cytoadherence.

IFA revealed punctate *Tv*PI4P5K staining in the T1 isolate cytoplasm slightly colocalized with the flagellar PIP$_2$, whereas the TH17 isolate had punctate *Tv*PI4P5K staining in the cytoplasm and plasma membrane, with partial colocalization with PIP$_2$ in the plasma membrane, perhaps related to PIP$_2$ production (Figs 2E and S3A). HA-*Tv*PI4P5K Wt and K136A (Fig 2F) were overexpressed five-fold in TH17 trophozoites (Fig 2G), increasing and depleting PIP$_2$ abundance respectively (Fig 2H), consistent with the IFA signals in the plasma membrane (Figs 2I and S3B), indicating that *Tv*PI4P5K is responsible for the plasma membrane PIP$_2$ production. However, K136A was detected in discrete dots in the cytoplasm, implying that PI4P5K kinase activity may have a role in cell membrane translocation. By density gradient ultracentrifugation on *T. vaginalis* lysates [36], the *Tv*PI4P5K signal was fractionated to the plasma membrane and various membrane compartments (S3C Fig), supporting the presence of *Tv*PI4P5K in the plasma membrane.

### PIP$_2$ hydrolysis in *T. vaginalis* flagellate-amoeboid transition and cytoadherence

Flagellate-amoeboid transition is critical for *T. vaginalis* cytoadherence [27,28]. TH17 flagellate trophozoites were cultured on a glass slide to trigger contact-dependent amoeboid morphogenesis (Fig 3A, S1 Movie) and after 30 minutes, ~80% of the TH17 non-transfectant or HA-*Tv*PI4P5K transfectant trophozoites transformed into the amoeboid form but only ~40% of the K136A mutant transformed (Fig 3B). When *T. vaginalis* trophozoites were co-incubated with *h*VECs, the cytoadherence activity was similar between the non-transgenic control and HA-*Tv*PI4P5K transfectant but reduced by 40% in the K136A mutant (Fig 3C). There was no change in the *Tv*PI4P5K overall signal intensity but the plasma membrane PIP$_2$ signal was depleted in the trophozoites 30 minutes post morphogenesis (Fig 3D). This was accompanied by reduced *Tv*PI4P5K in the cell membrane (Figs 3D, S4A and S4B) and increased IP$_3$ (S5 Fig), linking the potential plasma membrane PIP$_2$ hydrolysis with parasite morphogenesis. The PLC inhibitor, Edelfosine (ET-18-O-CH3, 1-octadecyl-2-O-methyl-glycero-3-phosphocholine), simultaneously maintained the plasma membrane PIP$_2$ level (Figs 3E and S4B), repressed IP$_3$ production (S5 Fig), and reduced amoeboid morphogenesis (Fig 3E). Similar results were observed in the dot blot of parasites undergoing amoeboid transformation (Fig 3F). The overall PIP$_2$ level reduced with morphogenesis was reversed by Edelfosine treatment. Likewise, the overall PIP$_2$ abundance in the HA-*Tv*PI4P5K transfectant was reduced upon morphogenesis but accumulated in the presence of Edelfosine. However, Edelfosine treatment had no effect in the PIP$_2$-depleted K136A mutant (S6 Fig). These results suggest that

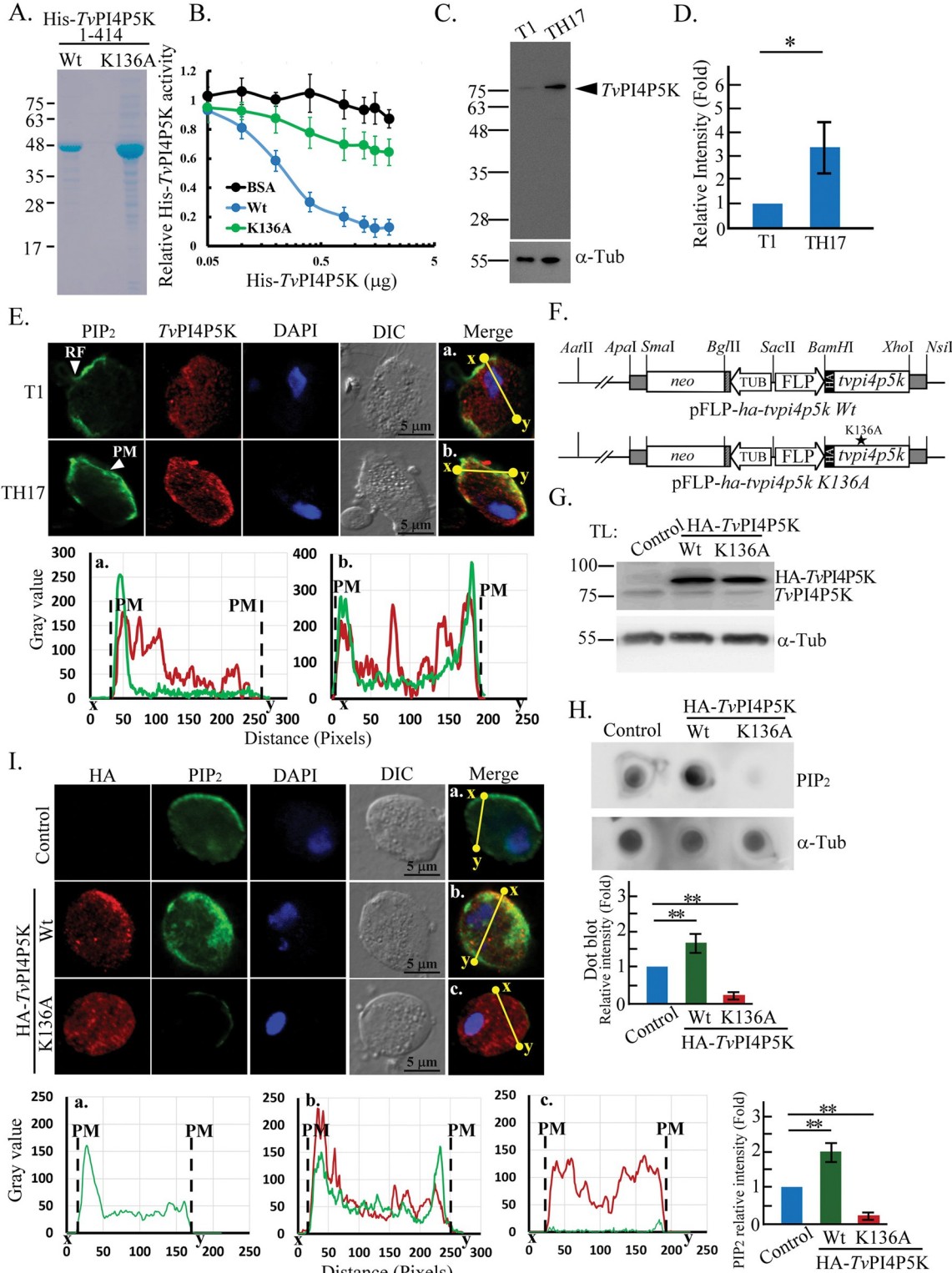

**Fig 2. *Tv*PI4P5K enzyme activity.** (A) The bacterial recombinant His-*Tv*PI4P5K wild-type kinase domain (Wt) and kinase-deficient mutant (K136A) were purified for various assays. (B) *Tv*PI4P5K *in vitro* kinase activity was evaluated using serially-diluted His-*Tv*PI4P5K, K136A mutant, or BSA control reacted with PI(4)P substrate in the presence of ATP. The relative activities of His-*Tv*PI4P5K Wt and K136A mutant were measured at different concentrations as shown in the plot (n = 3, mean ± SD). (C) The total lysates (TL) from T1 and TH17 isolates were subjected to western blotting for *Tv*PI4P5K and α-tubulin detection. The leftward

arrowhead indicates the *Tv*PI4P5K signal. The relative *Tv*PI4P5K signal intensity was quantified as shown in (D). (E) T1 and TH17 flagellate trophozoites were double-stained with anti-PIP$_2$ or anti-*Tv*PI4P5K antibodies for IFA. Signal intensity distribution on the yellow line between x and y sites in the representative micrograph (a, b) was analyzed by ImageJ as shown in the plot (a, b). RF indicates recurrent flagellum and PM indicates the plasma membrane boundary. (F) Plasmids with *ha-tvpi4p5k* Wt or K136A mutant driven by the Fibronectin-like protein (FLP) promoter and neomycin selection marker driven by the α-tubulin promoter, were constructed to overexpress HA-*Tv*PI4P5K and the derived mutant. (G) The protein lysates from the non-transgenic control or HA-*Tv*PI4P5K Wt and K136A transfectants were detected by western blotting using anti-*Tv*PI4P5K or anti-α-tubulin antibody. (H) The protein lysates from (G) were subjected to a dot blot assay to detect PIP$_2$ and α-tubulin. The assay was processed in three biological repeats, and the relative intensities of PIP$_2$ normalized to α-tubulin were quantified as shown in the bar graph (n = 3, mean ± SD). (I) The parasites without or with HA-*Tv*PI4P5K Wt and K136A overexpression were fixed and double-stained with anti-PIP$_2$ and anti-HA antibodies for IFA. Signal intensity distribution on the yellow line between x and y sites in the representative micrograph (a-c) was analyzed as shown in the plot (a-c). PM indicates the plasma membrane boundary. The assay was processed in three biological repeats, and the relative intensities of the overall PIP$_2$ signal were quantified as shown in the bar graph. (n = 3, mean ± SD). For (H) and (I), significant differences were analyzed by Student's t-tests, with $p < 0.05$(*) and $p < 0.01$ (**).

*T. vaginalis* morphogenesis may involve PLC-dependent plasma membrane PIP$_2$ degradation. Meanwhile, Edelfosine treatment did not affect parasite vitality, proving its inhibitory efficacy (S7 Fig).

### PIP$_2$ signaling regulates *T. vaginalis* intracellular calcium

To validate whether PIP$_2$ hydrolysis influences the *T. vaginalis* intracellular calcium, a calcium indicator, Calcium Green-1-AM (CG) was preloaded into the parasite for microscopic fluorescence detection. The CG signal intensity was higher in the TH17 adherent isolate (Fig 4A) and further increased in the PIP$_2$-overexpressed HA-*Tv*PI4P5K transfectant but decreased in the PIP$_2$-depleted K136A mutant (Fig 4B), indicating that *T. vaginalis* PIP$_2$ level was positively associated with the intracellular calcium content. When the TH17 flagellates were cultured on a glass slide, the CG signal was increased in the trophozoites 30-min post-incubation but reduced by Edelfosine treatment and abolished by BAPTA-AM, a cell-permeant calcium chelator can be used to control intracellular calcium level (Fig 4C, S2 Movie). Edelfosine and BAPTA-AM treatment also suppressed parasite morphogenesis (Fig 4D) and cytoadherence (Fig 4E). Simultaneously, the CG signal increased upon morphogenesis and the proportion of amoeboid form reduced in the presence of EGTA, an extracellular calcium chelator (S2 Movie, S8 Fig), suggesting that extracellular calcium influx may be required for *T. vaginalis* flagellate-amoeboid transition. In summary, the PIP$_2$ hydrolysis followed by increased intracellular calcium is vital for parasite amoeboid morphogenesis and cytoadherence. Edelfosine, BAPTA-AM, or EGTA treatment did not affect parasite viability (S7 Fig), supporting their inhibitory effects.

### PIP$_2$ signaling modulates actin organization in *T. vaginalis*

Previously, we demonstrated that actin polymerization is crucial in *T. vaginalis* flagellate-amoeboid transition [27,28], so we further examined whether PIP$_2$ signaling regulates amoeboid morphogenesis through actin cytoskeleton organization. The transfectants equally over-expressing HA-*Tv*PI4P5K Wt and K136A (Fig 5A) were fractionated into the supernatant (G-actin) and pellet (F-actin) to evaluate actin polymerization. Compared to the non-transgenic control parasite, the F-actin content was unchanged in HA-*Tv*PI4P5K Wt with PIP2 overexpression (Fig 5B) but reduced in the PIP$_2$-depleted K136A mutant (Fig 5C), revealing that PIP$_2$ may be involved in actin polymerization. Moreover, F-actin polymerization was inhibited in the parasites treated with Edelfosine (Fig 5D) or BAPTA-AM (Fig 5E). Conversely, a calcium ionophore, A23187, slightly increased CG signal intensity and the F-actin ratio in the

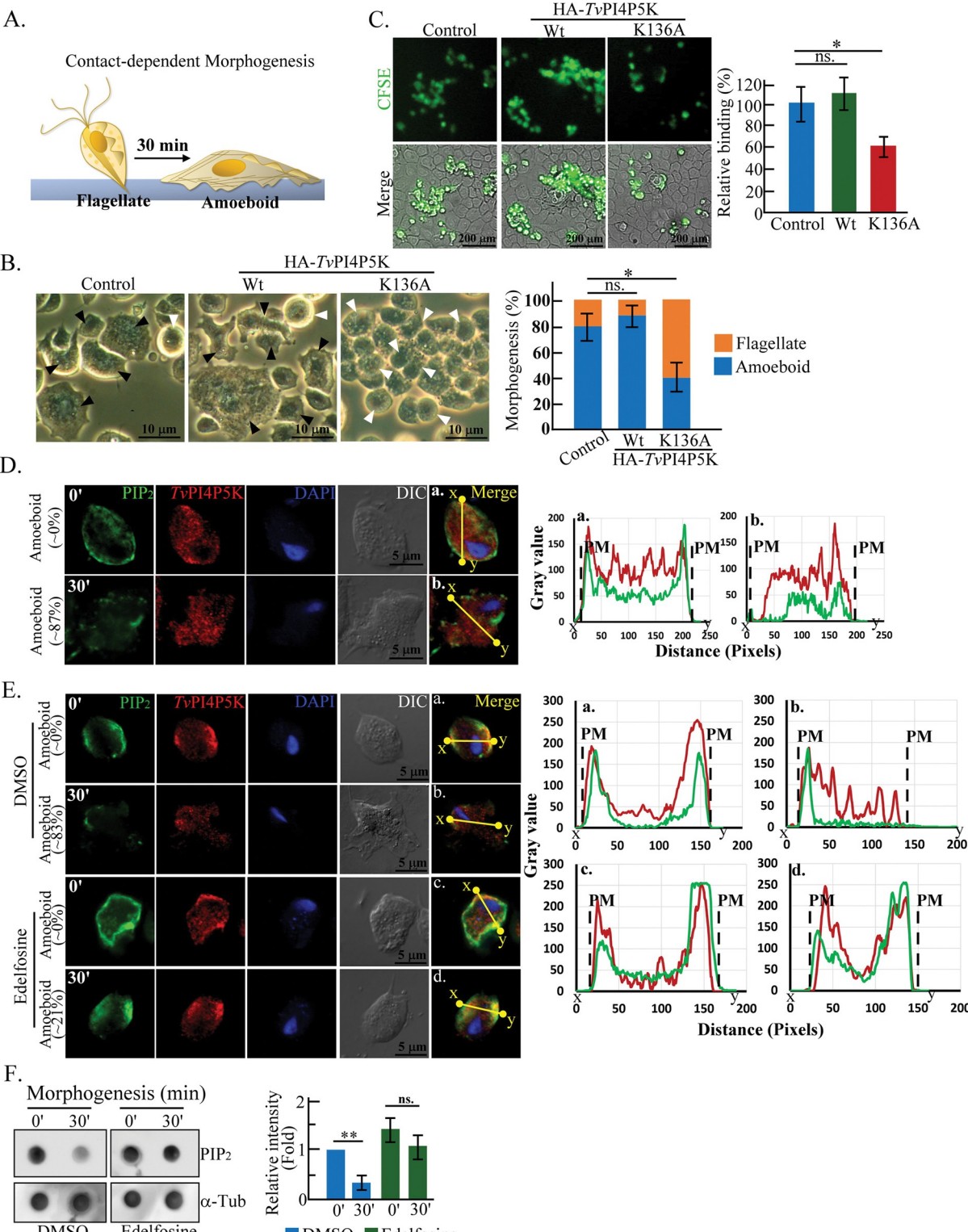

**Fig 3. PIP$_2$ hydrolysis in *T. vaginalis* flagellate-amoeboid transition and cytoadherence.** (A) Culturing the parasite on a glass slide for 30 min at 37˚C triggers the flagellate-amoeboid transition of TH17 trophozoites in contact with a solid surface, referred to as contact-dependent morphogenesis. (B) The non-transgenic TH17 control and the transfectants overexpressing HA-*Tv*PI4P5K Wt and K136A mutant were cultured in a T25 flask for 30 min to observe the morphology by phase-contrast microscopy. Black and white arrowheads indicate amoeboid and flagellate trophozoites, respectively. The assay was processed in three biological repeats to measure the proportion of the flagellate versus

amoeboid trophozoites, as shown in the bar graph (n = 3, mean ± SD). (C) The CFSE-prelabeled parasites were co-cultured with *h*VECs for 1 hr and bound parasites were detected by a confocal microscope. The assay was processed in three biological repeats, and the relative ratio of the bound parasite was quantified as shown in the bar graph when the non-transgenic control was defined as 100% (n = 3, mean ± SD). (D) The TH17 free flagellate trophozoites before (0′) and after culture on a glass slide for 30 min (30′) were fixed for IFA double staining with anti-PIP₂ and anti-*Tv*PI4P5K antibodies. The fluorescence signal was observed by confocal microscopy, and morphology was recorded in the DIC mode to estimate the amoeboid trophozoite ratio, as shown in the parentheses. (E) The TH17 flagellate trophozoites pretreated with DMSO or Edelfosine before (0′) and after adherence to a glass slide for 30 min (30′) were fixed for IFA double-staining with anti-PIP₂ and anti-*Tv*PI4P5K antibodies. For (D.) and (E.), signal intensity distribution on the yellow line between x and y sites in the representative micrograph (a, b of D. and a-d of E.) was analyzed by ImageJ as shown in the corresponding plot (a, b of D. and a-d of E.). PM indicates the plasma membrane boundary. (F) The protein lysates extracted from the trophozoites with or without Edelfosine treatment before (0′) and after culture in a T25 flask for 30 min (30′) were subjected to dot blot to detect PIP₂ and α-tubulin. The assay was processed in three biological repeats, and the relative PIP₂ signal intensity normalized to α-tubulin is shown in the bar graph (n = 3, mean ± SD). For (B), (C), and (F), the significant differences for the paired samples were analyzed by Student's t-tests, with $p < 0.05$(*), $p < 0.01$(**), ns. no significant difference.

parasite (S9 Fig), suggesting that intracellular calcium triggered from PIP₂ hydrolysis is required for actin polymerization.

## Iron is involved in *T. vaginalis* PIP₂ signaling

Iron was previously found to transiently trigger PKA-dependent signaling to regulate the nuclear translocation of the Myb3 transcription factor in *T. vaginalis* [17], so we investigated whether iron also regulates PIP₂ signaling. TH17 trophozoites were cultured overnight in the normal, iron-depleted, and iron-replete growth media for analysis, revealing that the PIP₂ (Fig 6A) and *Tv*PI4P5K (Fig 6B) levels were similar in the normal-iron or iron-replete parasites but higher than in the parasites depleted of iron, implying that the iron in the normal growth medium sufficiently affects PIP₂ and *Tv*PI4P5K expression in *T. vaginalis*. IFA double-staining showed that the plasma membrane PIP₂ signal intensity was similar in the normal- and iron-replete parasites and more robust than in the iron-depleted parasites (Fig 6C). Punctate *Tv*PI4P5K staining was observed in the cytoplasm of iron-depleted parasites with a lower level of PIP₂, whereas numerous *Tv*PI4P5K dots around the cell membrane were partially colocalized with a higher amount of PIP₂ in the plasma membrane in the normal- and iron-replete parasites (Figs 6C, S10A and S10B). Iron may increase *Tv*PI4P5K plasma membrane localization for PIP₂ production. Since iron did not affect *tvpi4p5k* gene transcription (S10C Fig), iron may modulate *Tv*PI4P5K expression translationally or post-translationally.

## Iron affects *T. vaginalis* actin-centric activities

Iron regulates *Tv*PI4P5K expression and plasma membrane localization, thus, we examined whether iron also affects downstream PIP₂ signaling cascades and cytoskeleton behaviors in *T vaginalis*. In contrast to the normal-iron parasites, the levels of intracellular CG signal (Fig 7A), amoeboid morphogenesis (Fig 7B), cytoadherence (Fig 7C), and F-actin polymerization (Fig 7D) were diminished in the iron-depleted parasites but partially restored to the basal level by A23187 challenge, suggesting the potential role of iron in intracellular calcium and actin-based cytoskeleton behaviors of *T. vaginalis*. However, A23187 had no significant effect on *Tv*PI4P5K plasma membrane translocation and expression or PIP₂ level in the iron-depleted TH17 trophozoites, indicating that A23187 only activates PIP₂ downstream signals (S11 Fig) and highlighting the significance of intracellular calcium in *T. vaginalis* cytoskeleton regulation.

## PIP₂ signaling in *T. vaginalis* cytopathogenic activity

*T. vaginalis* could lyse multiple types of human cells with contact-dependent disruption or soluble factors [5,10,11]. To clarify the roles of iron and PIP₂ signaling in *T. vaginalis* cytotoxicity,

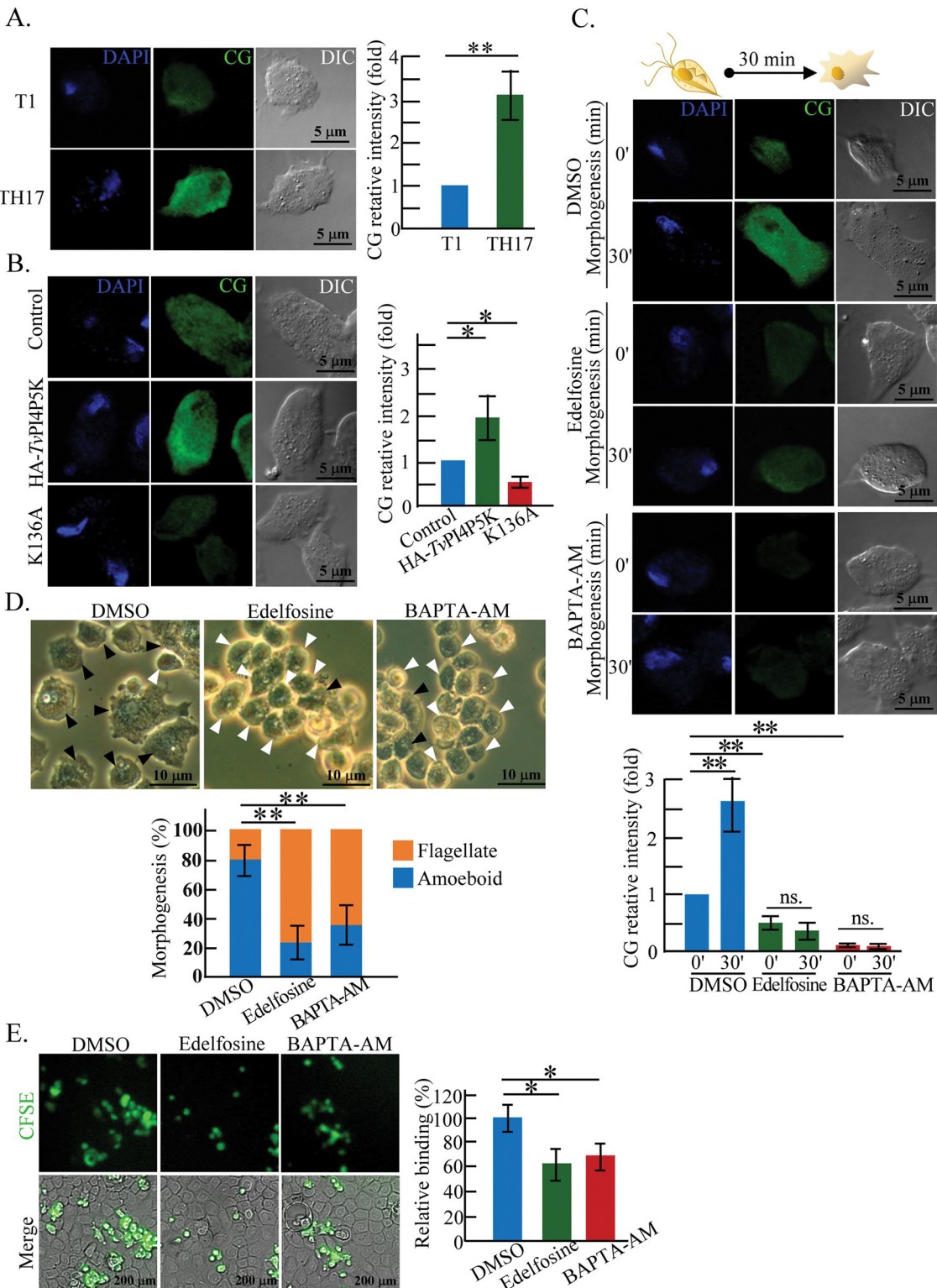

**Fig 4. PIP$_2$ signaling regulates *T. vaginalis* intracellular calcium.** (A) The T1 and TH17 trophozoites preloaded with Calcium Green (CG) fluorescence dye were fixed for detection by a confocal microscope. (B) The CG-preloaded TH17 non-transgenic flagellate trophozoites and those overexpressing HA-*Tv*PI4P5K Wt and K136A mutant were fixed for confocal microscopic detection. (C) The TH17 flagellates pretreated with DMSO, Edelfosine, and BAPTA-AM were loaded with CG. The trophozoites, before (0′) and after culture on a glass slide for 30 min (30′), were fixed for confocal microscopy. The assay was processed in three

biological repeats, and the relative intensity of the CG signal was quantified in 300 trophozoites from five independent microscopic fields as shown in the bar graph (n = 3, mean ± SD). (D) The TH17 flagellates pretreated with DMSO, Edelfosine, and BAPTA-AM were cultured in a T25 flask for 30 min to record the parasite morphology using a phase-contrast microscope. Black and white arrowheads indicate amoeboid and flagellate trophozoites, respectively. The proportion of flagellate versus amoeboid form was quantified in 300 trophozoites within five independent microscopic fields, as shown in the bar graph. The assay was processed in three biological repeats (n = 3, mean ± SD). (E) CFSE-prelabeled TH17 flagellate trophozoites pretreated with DMSO, Edelfosine, or BAPTA-AM were co-cultured with *h*VECs for 1 hr. After washing, the bound parasites on *h*VECs were fixed for CFSE signal detection by confocal microscopy. The assay was processed in three biological repeats, and the relative ratio of bound trophozoites was calculated as shown in the bar graph when the DMSO control was defined as 100% (n = 3, mean ± SD). Significant differences were statistically analyzed by Student's t-tests, with *p* < 0.05(*) and *p* < 0.01(**).

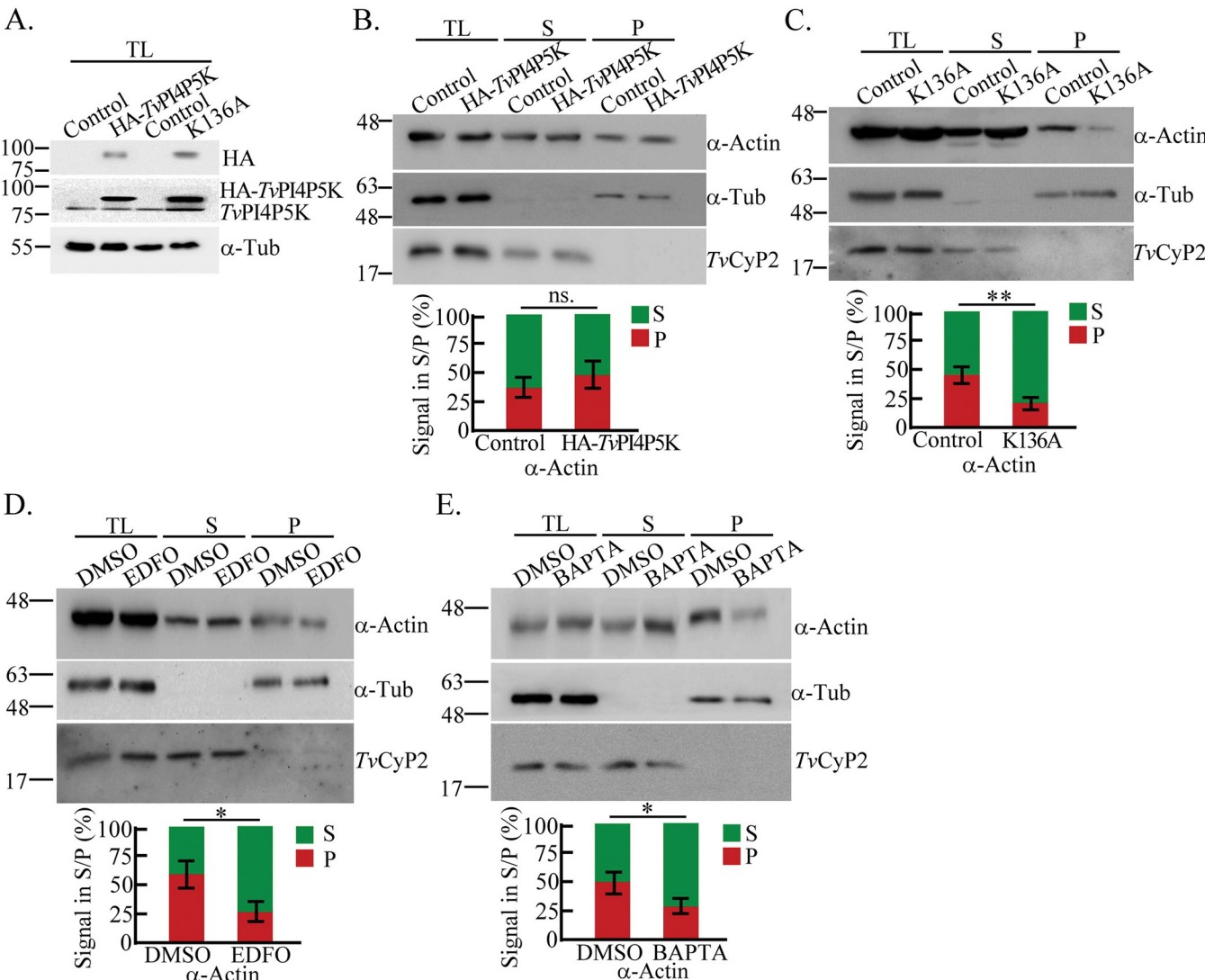

**Fig 5. PIP$_2$ signaling modulates actin organization in *T. vaginalis*.** (A) The total lysates from the non-transgenic TH17 trophozoites and those overexpressing HA-*Tv*PI4P5K Wt and K136A were subjected to western blotting detection by anti-HA anti-*Tv*PI4P5K, and anti-α-tubulin antibodies. The total lysates from the TH17 transfectant overexpressing HA-*Tv*PI4P5K Wt (B), K136A mutant (C), or the non-transgenic TH17 parasites treated with DMSO, Edelfosine (EDFO) (D), and BAPTA-AM (E), were fractionated into G-actin containing supernatant (S) and F-actin containing pellet (P) fractions for western blotting. The assay was processed in three biological repeats. The ratio of α-actin signals in the supernatant versus pellet fraction was quantified, as shown in the bar graphs. (n = 3, mean ± SD). Significant differences were statistically measured by Student's t-tests, with *p* < 0.05(*), *p* < 0.01(**), and ns, no significant difference.

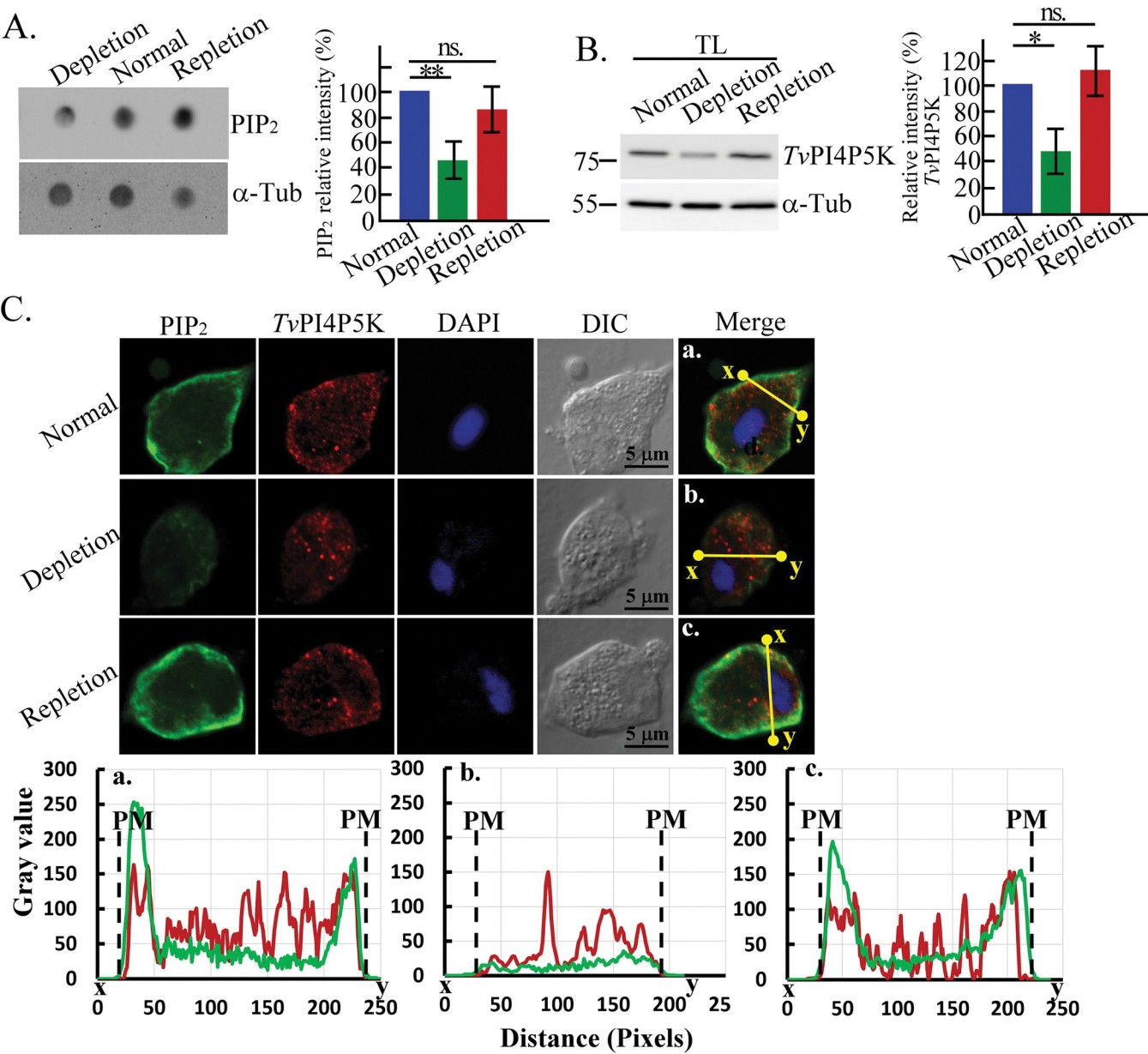

**Fig 6. Iron is involved in *T. vaginalis* PIP$_2$ signaling.** The TH17 trophozoites were incubated in normal-iron, iron-depleted, and iron-repletion medium overnight for further assays. (A) The protein lysates extracted from the trophozoites were subjected to dot blot assay with anti-PIP$_2$ and α-tubulin detection. The blot assay was processed in three biological repeats, and the relative signal intensity of PIP$_2$ normalized to α-tubulin is shown in the bar graph (n = 3, mean ± SD). (B) The total protein lysates (TL) from TH17 trophozoites under various iron conditions were sampled for western blotting using antibodies as indicated. The assay was processed in three biological repeats, and the relative signal intensity is shown in the bar graph (n = 3, mean ± SD). For (A) and (B), significant differences for the paired samples were analyzed by Student's t-tests, with $p < 0.05$(*), $p < 0.01$(**), ns. no significant difference. (C) The TH17 flagellate trophozoites at different iron conditions were fixed for IFA double staining with anti-PIP$_2$ and anti-*Tv*PI4P5K antibodies. The signal intensity distribution on the yellow line between the x and y sites in the representative micrograph (a-c) was analyzed by ImageJ, as shown in the corresponding plots (a-c). PM indicates the plasma membrane boundary.

the parasites precultured with or without iron were inoculated into *h*VECs at different multiplicity of infection (MOI) for the lactate dehydrogenase (LDH) cytotoxicity assay. Since the parasites started to die after 5 hours of culture in keratinocyte serum-free medium under 5% CO$_2$ (S12A Fig), we monitored the early 1-hour cytotoxicity to rule out the influences from the

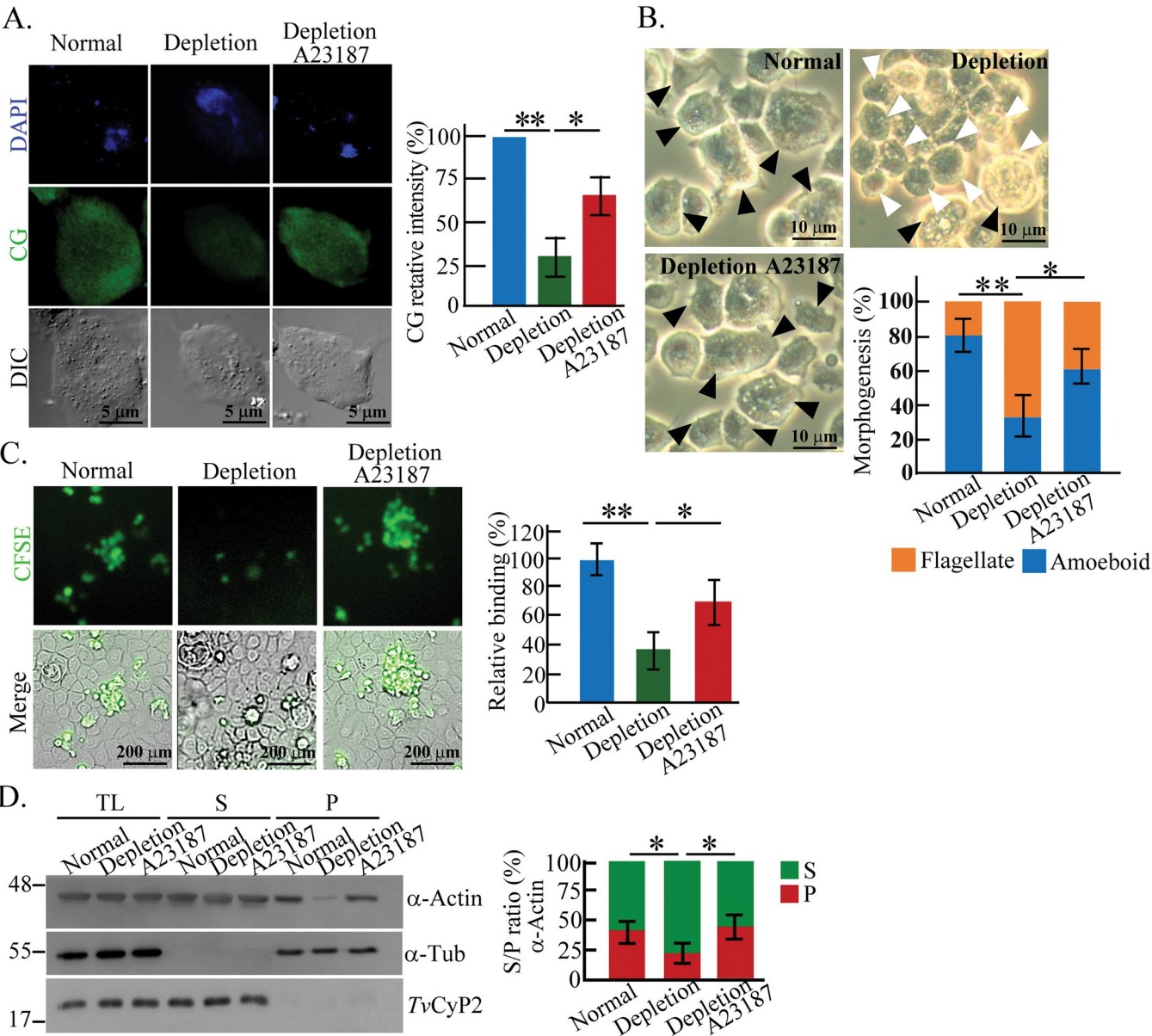

**Fig 7. Iron affects *T. vaginalis* actin-centric activities.** TH17 trophozoites were cultured in normal-iron and iron-depleted medium or treated with A23187. The parasites were sampled for CG detection (A), morphogenesis analysis (B), cytoadherence assay (C), and F-actin fractionation (D). For CG detection, the relative signal intensity averaged from 300 trophozoites within five independent microscopic fields was shown in the bar graph. For the morphogenesis assay, the flagellates were cultured in a T25 flask for 30 min to record the parasite morphology using a phase-contrast microscope. Black and white arrowheads indicate amoeboid and flagellate trophozoite, respectively. The proportion of flagellate versus amoeboid form was quantified in 300 trophozoites within five independent microscopic fields, as shown in the bar graph. For cytoadherence assay, the CFSE-labeled trophozoites were co-cultured with *h*VECs for 1 hr. After washing to remove unbound trophozoites, the bound trophozoites were fixed for CFSE detection, and the relative cytoadherence capacities between samples were measured as shown in the bar graph. For G/F-actin fractionation, the lysates from the trophozoites were fractionated into G-actin-containing supernatant and F-actin-containing pellet fractions for western blotting. The ratio of α-actin signals in the supernatant (S) versus pellet fractions (P) was quantified as shown in the bar graph. All assays were processed in three biological repeats (n = 3, mean ± SD). The significant differences are statistically measured by Student's t-test, with $p<0.05$(*), $p<0.01$(**), and ns, no significant difference.

assay, showing that the overall cytotoxicity increased with increasing MOI. In normal-iron parasites at MOI over 4, the cytotoxicity slightly induced in the PIP$_2$-overexpressed HA-*Tv*PI4P5K Wt was reduced in the K136A mutant with PIP$_2$ depletion and those treated with

Edelfosine or BAPTA-AM and co-cultured with the protease inhibitor to a similar extent as the iron-depleted parasites (Fig 8A). Additionally, the cytotoxicity of iron-depleted parasites with A23187 activation was restored to the normal-iron parasite level (Fig 8A). In a 4-hr assay, the parasites caused similar cytotoxic tendencies with extents slightly greater (S12B Fig). However, the LDH activity remained low in the spent medium with the parasites only, ruling out the LDH leakage from the parasites in the 1-hr cytotoxicity assay (S12C Fig). PIC did not directly affect *h*VECs viability (S12D Fig). Regarding parasite contact-dependent cytolysis, the *h*VECs lysis area per trophozoite for the normal-iron parasites was slightly changed in the HA-*Tv*PI4P5K Wt transfectant, whereas the areas lysed by the K136A mutant and the parasites pretreated with Edelfosine and BAPTA-AM or co-cultured with protease inhibitors were reduced to that like the iron-depleted parasites. Again, the reduced cytolytic area in the parasites with iron depletion was recovered by A23187 to the normal-iron parasite level, supporting the crucial roles of iron and PIP$_2$ signaling cascades in contact-dependent cytolysis (Fig 8B). Notably, the overall parasite cytotoxicity and cytolysis activities were also suppressed by Latrunculin B (LatB), an F-actin assembly inhibitor [28] (Fig 8A and 8B), revealing that the actin cytoskeleton likely contributes to *T. vaginalis* cytolytic activity. Together, the PIP$_2$-signaling-cascade-mediated actin cytoskeleton may be involved in the contact-dependent cytotoxic effects, associated with extracellular protease activities. Environmental iron is required for *T. vaginalis* intact cytotoxicity, but whether PIP$_2$ regulates protease surface expression on or release from *T. vaginalis* needs further elucidation.

## Conclusion

Specific *Tv*PI4P5K and PIP$_2$ expression was observed in the *T. vaginalis* adherent isolate. During parasite flagellate-amoeboid transition, cell membrane PIP$_2$ was cleaved by a PLC-dependent pathway, increasing intracellular calcium essential for cytoskeleton activities, including actin remodeling, morphogenesis, and cytoadherence, which could be inhibited by Edelfosine or BAPTA and activated by A23187. When the iron content in the normal growth medium sufficiently elicited PIP$_2$ signaling, iron simultaneously regulated the expression and plasma membrane localization of *Tv*PI4P5K for PIP$_2$ production. In the host-parasite interaction, PIP$_2$ signaling cascades modulated parasite morphogenesis and cytoadherence, contributing to the cytopathic effects by extracellular protease-associated cytolysis in an actin cytoskeleton-dependent manner (Fig 9). The PIP$_2$-triggered extracellular cytopathic effectors warrant continued identification but this study provides an insight into a new dimension of *T. vaginalis* pathogenicity.

## Discussion

### Plasma membrane PIP$_2$

Since PIP$_3$ is undetectable in *T. vaginalis*, PIP$_2$ may be a significant plasma membrane signaling moiety in this parasite. In mammalian cells, PIP$_2$ is localized to lipid rafts, selectively regulating different cellular responses [37]. In *Entamoeba histolytica*, cholesterol stimulates PIP$_2$ enrichment in uroid lipid rafts at the trailing edge of a motile polarized trophozoite, increasing intracellular calcium concomitant with increased motility [38]. In T-cell signaling, plasma membrane PIP$_2$ recruits the activated ezrin-radixin-moesin (ERM) binding, spatially regulating actin polymerization. The deactivated ERM is released from the plasma membrane into the cytoplasm, thereby changing membrane fluidity and reshaping the membrane structure necessary for adhesion [39]. In the same way, reduced PIP$_2$ in the plasma membrane of *T. vaginalis* amoeboid trophozoites may facilitate the membrane reshaping required for morphogenesis.

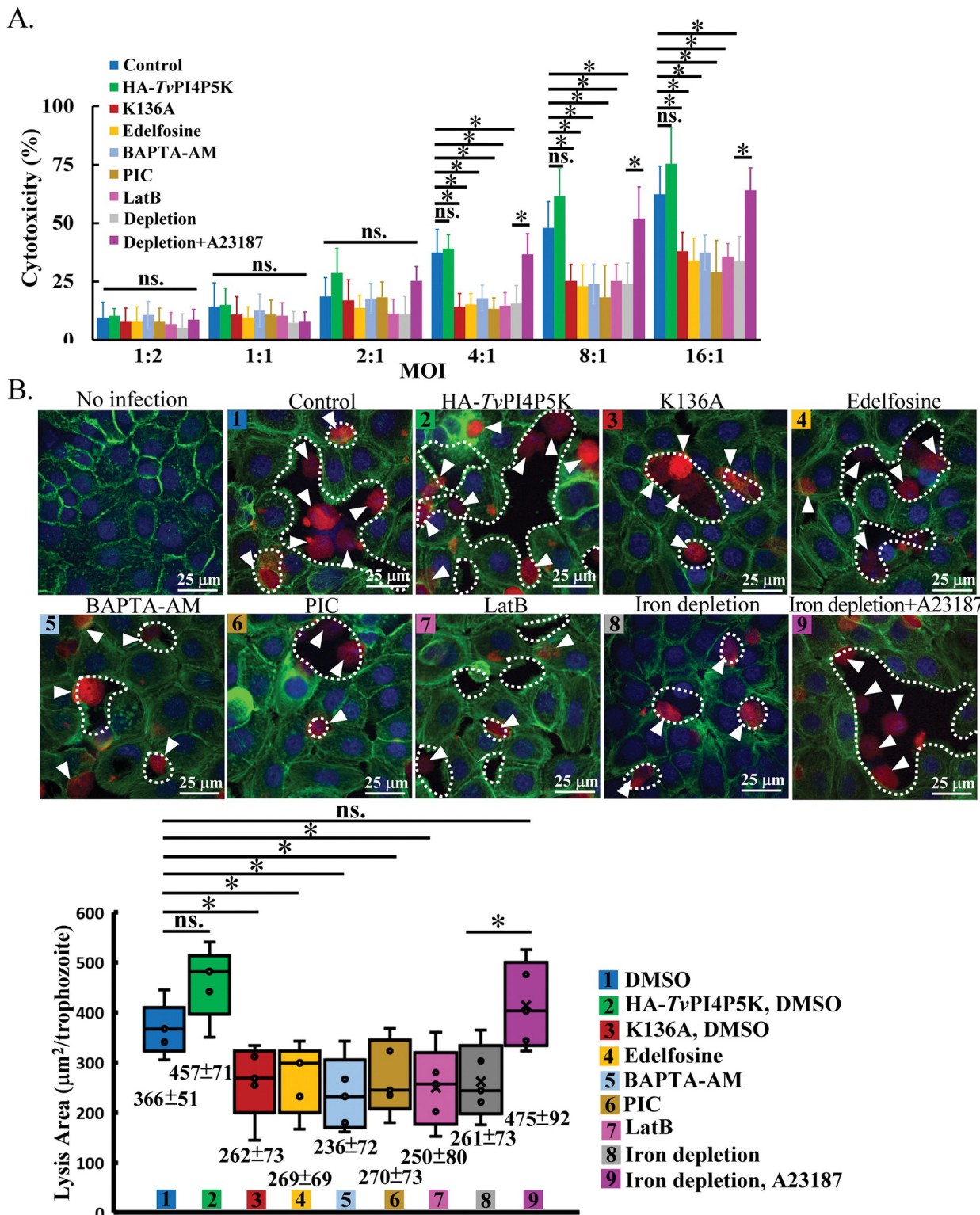

**Fig 8. PIP$_2$ signaling in *T. vaginalis* cytopathogenic activity.** The TH17 trophozoites without (Control) or with HA-*Tv*PI4P5K Wt or K136A overexpression pretreated with DMSO, Edelfosine, BAPTA-AM, or LatB were inoculated into *h*VECs in the presence or absence of a protease inhibitor cocktail (PIC). (A) The parasites were co-cultured with *h*VECs at different MOI for 1 hr in the LDH cytotoxicity assay. The assay was performed in three biological repeats to measure the relative cytotoxicity (%) as shown in the bar graph (n = 3, mean ± SD). (B) The trophozoites labeled with CMRA were co-cultured with *h*VECs (MOI = 4) for 1 hr. All specimens were fixed for staining with FITC-conjugated phalloidin.

Nuclei were stained with DAPI. The white dashed line indicates the area lysed by the parasite on the *h*VECs monolayer. The assay was performed in three biological repeats, and the average lysis area per trophozoite is shown in the Box-whisker plot (n = 5, mean ± SD). The significant differences of grouped samples were analyzed by Student's t-tests with the $p < 0.05$(*), $p < 0.01$(**), and ns, no significant difference.

In the higher eukaryotic organism, PIP$_2$ is cleaved through a well-defined PLC hydrolysis into IP$_3$ and DAG [40] to release intracellular calcium [41] and activated PKC [42], respectively. Also, in mammal sperm cells, PKC regulates F-actin formation [43], and IP$_3$ involves intracellular calcium regulation [44]. Our data showed that PLC-dependent PIP$_2$ cleavage might be conserved in *T. vaginalis* (Fig 3). The downstream intracellular calcium cascade is essential to F-actin assembly and the extended behaviors, crucial for parasite host colonization [27,28]. However, whether the plasma membrane PIP$_2$ can manipulate peripheral actin polymerization directly or indirectly through a calcium second messenger warrants further investigation. The TrichDB database (https://trichdb.org/trichdb/app) [45] contains one potential phospholipase C precursor (TVAG_180400) with low sequence consensus to mouse PLCZ1 (Q8K4D7) and numerous calcium channels, transporters, and binding proteins, but the biological nature of *T. vaginalis* PLC remains to be characterized. IP$_3$ receptors are unusual among endoplasmic reticulum proteins functionally expressed at the PM, where very few IP$_3$ receptors contribute to calcium entry [46]. The increasing intracellular calcium level in the transforming *T. vaginalis* could be effectively reduced in the continuous presence of EGTA (S8 Fig), suggesting that *T. vaginalis* PIP$_2$ signaling modulates extracellular calcium influx. The detail of how PIP$_2$ regulates calcium flow in *T. vaginalis* is an intriguing question to be addressed. Although the adherent isolates in this study expressed higher *Tv*PI4P5K and PIP$_2$

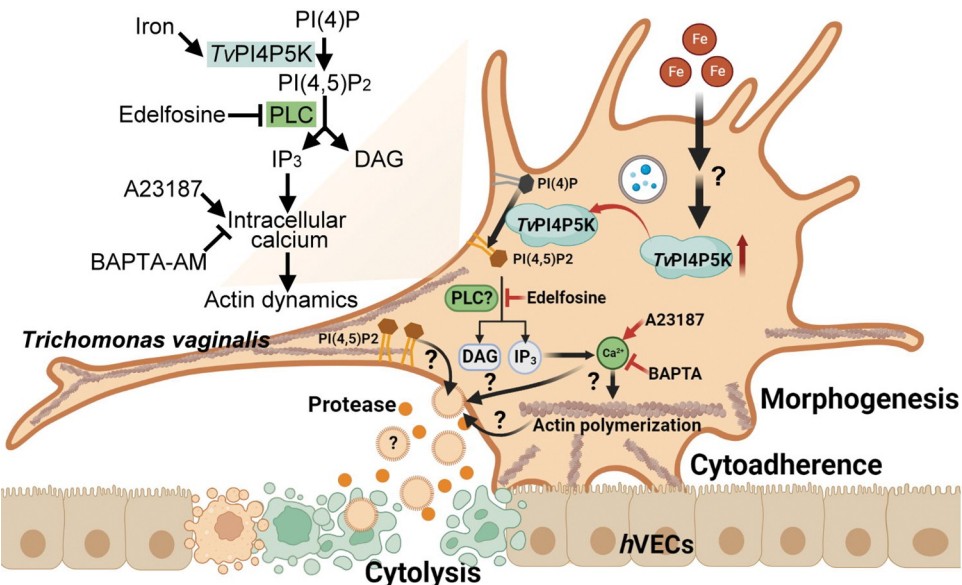

**Fig 9. PIP$_2$ signal transduction in *T. vaginalis* pathogenesis.** *Tv*PI4P5K phosphorylates PI(4)P into PI(4,5)P$_2$, which is cleaved by PLC hydrolysis. Although DAG has not yet been identified in *T. vaginalis*, the induced intracellular calcium regulates actin dynamics essential for morphogenesis and cytoadherence. The cytoskeleton activities were also activated and repressed by A23187 and BAPTA-AM, respectively, indicating the role of calcium in cytoskeleton regulation. The details of how environmental iron induces *Tv*PI4P5K expression and trafficking to the plasma membrane for PIP$_2$ production remain unexplored, but the cell membrane PIP$_2$ signaling cascades are involved in host cell cytolysis in a protease-dependent manner. However, whether PIP$_2$, the actin cytoskeleton, and calcium individually or coordinately regulate *T. vaginalis* cytotoxicity requires further elucidation. (The illustration was created with BioRender.com).

(Fig 1), the amoeboid morphogenesis and cytoadherence activities were not significantly changed in the less adherent isolates overexpressing *Tv*PI4P5K (S13 Fig), suggesting that PIP$_2$ might be an effector essential but not sufficient for evoking *T. vaginalis* actin-based capabilities, and the less adherent isolates may have evolved a divergent regulatory mechanism.

## Flagellar PIP$_2$ of *T. vaginalis*

In many postmitotic cells, the PIP$_2$ level determines the primary cilia length, and shortening from ciliary fission is attributed to constitutively increased ciliary PIP$_2$ and F-actin polymerization [47]. Also, the gelsolin-severing F-actin filaments inhibit flagellar motility in guinea pig sperm [48]. Therefore, whether the specified PIP$_2$ in recurrent flagellar of the less adherent isolate regulates actin activities leading to flagellation or the flagellar locomotion, such as motor stirring driving force, are intriguing questions worthy of further investigation. In contrast to some calcium-activated actin effectors severing the actin meshwork in the higher eukaryotic organisms [32], the increased intracellular calcium promoted F-actin assembly and consequent cytoskeleton behaviors in the amoeboid *T. vaginalis* trophozoite (Figs 5 and S9), indicating a different calcium function in *T. vaginalis* actin regulation. Intracellular calcium signaling is essential for the apicomplexan parasites in cell motility, invasion, and egress of host cells [49,50]. However, how upstream signaling mediates the calcium concentration remains unknown in these microorganisms.

## *Tv*PI4P5K cell membrane trafficking

Previous studies have reported that mouse Arf6 resides on the endosome and plasma membrane to regulate protein trafficking between the compartments [51,52] and PIP5K endosome-plasma membrane trafficking [53]. Punctate *Tv*PI4P5K staining in the cytoplasm and approaching the plasma membrane in the presence of iron (Figs 6C and S10) suggest that *Tv*PI4P5K may be enclosed in or reside on particular membrane-bound vesicles. Meanwhile, multiple Arf-like proteins found in the TrichDB database supported that *Tv*PI4P5K plasma membrane trafficking mediated with iron is likely through a similar protein cargo delivery system [45].

## Iron and signal transduction

Regarding the host-parasite interactions, extracellular calcium may influence trichomonad recognition and binding fibronectin [54] and enhance *Tv*CLP-mediated *T. vaginalis* aggregation on the host cells [4]. Our study further expands the role of calcium to function as an intracellular second messenger contributing to *T. vaginalis* pathogenesis.

Iron is an essential nutrient for *T. vaginalis* and is acquired via a highly specific receptor-mediated mechanism from the host [55]. The absence of iron in the culture medium decreases cell growth and triggers morphological changes from the amoeboid to a rounded flagellate form by an unknown mechanism. Iron exerts versatile roles in parasite growth and virulence, including the expression of genes required for metabolism [19] or virulence [21,22]. Also, iron increases *T. vaginalis* resistance to complement lysis for evading host immunity [23]. Iron may modulate histone acetylation at H3K27 and tri-methylation at H3K4 within the coding region of iron-responsive *ap65-1* and *pfo* genes, epigenetically regulating cytoadherence [16]. Meanwhile, iron promotes cAMP-dependent *Tv*PKAc activation to phosphorylate the Myb3 transcription factor for rapid nuclear import required for regulating *ap65-1* gene expression [17]. In bovine sperm cells, the high PKA activity elicits PIP$_2$ production via PI3K-associated PI4P5K activation, resulting in PLD activation and actin assembly [56]. More investigation is

required to elucidate the crosstalk of *Tv*PKAc to *Tv*PI4P5K activity and PIP$_2$ production in *T. vaginalis*.

Although HA-*Tv*PI4P5K overexpression promoted intracellular PIP$_2$ (Fig 2H), it did not significantly activate actin assembly (Fig 5) or consequent morphogenesis and cytoadherence activities beyond the basal level in the control parasites (Fig 3), implying that the restricted plasma membrane PIP$_2$ pool was adequate for *T. vaginalis* downstream regulation.

### Cytotoxicity

*T. vaginalis* actin-based phagocytosis damaged and phagocytosed host epithelium cells [9,10,11]. Therefore, PIP$_2$ signaling modulated actin dynamics, and contact-dependent cytolysis may be relevant (Fig 8). Moreover, contact-dependent cytolysis may require extracellular proteases to destroy the host cell membrane integrity. Intriguingly, *T. vaginalis* has been reported to export cysteine protease via a nonconventional lysosomal pathway [15] or express rhomboid protease on its surface [5]. Given that PIP$_2$ regulates endosomal trafficking and vesicle fusion between intracellular membrane compartments [57], PIP$_2$ signaling cascades may mediate parasite lysosome plasma membrane trafficking to an extracellular dump or present protease on the surface to damage host cells [5]. Furthermore, some specified proteins [14,58] or extracellular vesicles [59,60,61] secreted by the parasites may be uptaken by the host cells to modify the physiology during host-parasite interactions. The intracellular PIP$_2$ and calcium elevations involve rat neuroendocrine cell exocytosis [53] and coordinate with the actin cytoskeleton to control vesicle transportation and secretion [62]. As an actin assembly inhibitor, LatB effectively reduces *T. vaginalis* F-actin dynamics [28], but whether the cytopathic effects are directly regulated by PIP$_2$ or indirectly attributed to the subsequent signal cascades or actin cytoskeleton warrants further validation. This mechanism may provide new drug targets to restrict *T. vaginalis* pathogenicity at the initial infection.

## Materials and methods

### Cell culture

*T. vaginalis* incubated in TYI medium with 10% bovine serum at 37°C was defined as the normal-iron condition unless specified in the text. For iron-depleted or iron-replete culture conditions, the trophozoites were cultured in TYI growth medium supplemented with 50 μM 2, 2′-dipyridyl, or 250 μM ferrous ammonium sulfate, respectively [17]. For drug treatment, the parasite culture was pretreated with 10 μM of Edelfosine (TargetMol), 20 μM of BAPTA-AM (Abcam), 20 μM of A23187 (Abcam), or 1 μM of Latrunculin B (Sigma-Aldrich) in TYI medium on rotation for 90 min at 37°C. After washing with the medium to remove extracellular drugs, the trophozoites were immediately suspended in the medium or buffer for further assay. EGTA (2 mM) was used to chelate extracellular calcium in the medium. The clinical NTU isolate series were obtained from symptomatic vaginitis patients and maintained in the TYI medium supplemented with 1000 U of penicillin, 1000 μg of streptomycin, and 2.5 μg/ml of amphotericin B, with daily passaging over two weeks to remove microbial contaminants. For *T. vaginalis* cryopreservation, $2\times 10^7$ trophozoites of the clinical isolates were suspended in 1 ml of TYI medium with 7.5% DMSO and kept in the freezing container filled with isopropanol and then placed in a -80°C freezer over 24 hr. All isolates were deposited in the biosafety chamber of our institute for research only (National Taiwan University, College of Medicine). In contrast to the T1, G3, and TH17 experimental isolates [27,28], the fresh clinical isolates cultured short-term for weeks rather than years may preserve more intrinsic virulence. Human vagina epithelium cell (*h*VECs) was cultured in keratinocyte serum-free medium (Gibco) at 37°C with 5% CO$_2$.

## Trypan blue exclusion assay

The parasites were stained with 0.4% trypan blue in PBS to evaluate the viability of the parasites with or without drug treatment. The percentage of viable cells was measured in 300 trophozoites within five independent microscopic fields as follows:

$$\frac{Number\ of\ total\ trophozoites - Number\ of\ blue\ trophozoites}{Number\ of\ total\ trophozoites} \times 100.$$

## Plasmid construct

The pFLP-HA-*Tv*CyP2 plasmid [36] driven by the promoter (-786 to +11) of Fibronectin-like protein-1 (FLP) [63] was used to overexpress HA-*Tv*PI4P5K in *T. vaginalis*. The full-length DNA sequence of the *tvpi4p5k* gene (TVAG_462290) was amplified from the genomic DNA of the TH17 trophozoites using the primer pair *Tv*PI4P5K-BamHI-5' and *Tv*PI4P5K-XhoI-3' (Table 1). The PCR product was gel-purified and subcloned into the pFLP-HA-*Tv*CyP2 backbone vector with BamHI and XhoI restriction enzyme sites to generate the pFLP-HA-*Tv*PI4P5K plasmid. The K136A mutation in the pFLP-HA-*Tv*PI4P5K plasmid was generated by a site-directed mutagenesis kit according to the manufacturer's instructions (Toyobo) using the primer pair *Tv*PI4P5K (K136A)-5' and *Tv*PI4P5K (K136A)-3' (Table 1) to produce the pFLP-HA-*Tv*PI4P5K (K136A) plasmid.

To produce the recombinant proteins of *Tv*PI4P5K kinase domain (1 to 414 amino acid) wild type and K136A. DNA fragments amplified from the pFLP-HA-*Tv*PI4P5K or pFLP-HA-*Tv*PI4P5K (K136A) template plasmids using the primer pair *Tv*PI4P5K-BamHI-5 and *Tv*PI4P5K-414-XhoI-3' (Table 1) were separated in a 1.5% agarose gel and recovered using the gel extraction kit. The purified DNA fragments pre-digested with BamHI/XhoI were ligated into the pET28a expression vector to generate the pET28-His-*Tv*PI4P5K (1–414) and pET28-His-*Tv*PI4P5K (1–414) (K136A) plasmids.

The plasmid DNA was transfected into *T. vaginalis* by electroporation (BTX D DNA delivery system) to establish stable transfectant clones selected by paromomycin (200 μg/ml).

**Table 1. The oligonucleotide primers used in this study.**

| Primer | Sequence (5′ to 3′) |
|---|---|
| pFLP-HA-*Tv*PI4P5K plasmid | |
| *Tv*PI4P5K-BamHI-5' | AAGGATCCATGTCTCGCTCCGAATATAGTGA |
| *Tv*PI4P5K-XhoI-3' | AACTCGAGTTACTCTTGATCTTCAGATTTTG |
| pFLP-HA-*Tv*PI4P5K (K136A) plasmid | |
| *Tv*PI4P5K (K136A)-5' | *GCT*ACTCAAACGAAAGATGAAATGAAA |
| *Tv*PI4P5K (K136A)-3' | AAT AACATATCGACCGTCCCAAGT |
| pET28-His-*Tv*PI4P5K (1–414) plasmid | |
| *Tv*PI4P5K-414-XhoI-3' | AACTCGAGTTATGGATCAACACATGACATCTC |
| *β-tubulin* (qPCR) | |
| Tub-qPCR-5' | AAATCGTTCACATCCAAGCTGG |
| Tub-qPCR-3' | ACAAGGATAGCACGTGGGA |
| *tvpi4p5k* (qPCR) | |
| *Tv*PI4P5K-qPCR-5' | GAAGACAAAGAAGCAGAAAGAGA |
| *Tv*PI4P5K-qPCR-3' | GGTTCAACTTCTTTCACCTTTTCAT |
| *tvpi4p5k-2* (qPCR) | |
| *Tv*PI4P5K-2-qPCR-5' | GAAATTTTGAAGAAAGGGAAGTC |
| *Tv*PI4P5K-2-qPCR-3' | AAACACTAAATATCCTTGTCAAGGA |

The sequences of the restriction enzymes are underlined and mutations are italicized.

## Recombinant protein production

The plasmid was transformed into *Escherichia coli* (*E. coli*, BL21) for recombinant protein production as previously described [17,36]. *E. coli* culture at OD$_{260}$ 0.6 was induced with 1 mM IPTG and then incubated at 30°C for 3 hrs. The bacteria were recovered by 3000× *g* centrifugation and washed once with PBS followed by sonication. The bacterial lysate was centrifuged at 15000× *g* at 4°C for 15 min to remove cell debris and insoluble proteins before purification using a 1-ml Ni-NTA column as suggested by the supplier (Qiagen).

## RT-PCR and qPCR

RNA was extracted from *T. vaginalis* with TRIZOL reagent (Invitrogen) and reversely transcribed by Superscript III transcriptase (Invitrogen) using Oligo (dT) to produce the first-strand cDNA.

For qPCR, the relative expressions of *tvpi4p5k* and *tvpi4p5k-2* genes normalized to *β-tubulin* were quantified from the first-strand cDNA by QuantStudio 5 Real-Time PCR System (ThermoFisher Scientific) using a qPCRBIO SyGreen Blue Mix Lo-ROX kit (PCRBIOSYSTEMS). The reaction was performed with an initial denaturation at 95°C for 3 min, followed by 40 cycles of 94°C for 3 sec and 40 s at 60°C each. The *tvpi4p5k*, *tvpi4p5k-2*, and *β-tubulin* genes were amplified by the primer pairs of *Tv*PI4P5K-qPCR-5' /*Tv*PI4P5K-qPCR-3', *Tv*PI4P5K-2-qPCR-5'/*Tv*PI4P5K-2-qPCR-3', and Tub-qPCR-5'/Tub-qPCR-3', respectively (Table 1).

## Calcium Green intracellular calcium detection

The parasites were incubated in PBS containing 1% BSA and 5 µM Calcium Green-1-AM (CG, Invitrogen) at 37°C for 20 min. After removing the excess dye by washing with 1 ml PBS once, the parasites were cultured on the glass slide at 37°C for the specified time frame. After washing away the unbound parasites with 37°C pre-warmed PBS, the sample was fixed with 4% formaldehyde, then air-dried and mounted in anti-fade medium with DAPI. The fluorescence signal was observed by confocal microscopy (Zeiss, LSM-780) with an excitation wavelength of 506 nm and an emission wavelength of 531 nm. The signal intensity of 300 trophozoites within five microscopic fields was measured by ImageJ v.1.53k software (National Institutes of Health). Dynamics of the intracellular CG signal intensity in parasites upon amoeboid morphogenesis was recorded by time-lapse fluorescence microscopy with a capturing rate of 40 sec per one frame for 20 min.

## Immunofluorescence assay

The parasite trophozoites adhering to the glass slides were fixed with 4% formaldehyde, followed by the permeabilization with 0.5% saponin in TBS. After triple washes with TBS, the samples were incubated in TBS with 10% goat serum at 37°C for 20 min. After three washes with TBS to remove previous reactants, the sample was incubated with primary mouse anti-PIP$_2$ (400×, Echelon Biosciences), mouse anti-PIP$_3$ (200×, Echelon Biosciences), mouse anti-HA (200×, Sigma-Aldrich), rat anti-HA (100×, Roche) and rabbit anti-*Tv*PI4P5K (400×, Lab-made) antibodies diluted in TBS containing 1% BSA, at 4°C overnight. Then, the sample was washed three times with TBS and reacted with FITC or Cy3-conjugated goat anti-mouse IgM or IgG secondary antibodies prepared in TBS containing 1% BSA, at 37°C for 1 hr. The slide was washed three times in TBS and then air-dried at room temperature for 20 min. The sample mounted in an anti-fade medium with DAPI (Vector Labs) was observed by a confocal

microscope (Objective: Plan-Apochromat 100/1.40 Oil Ph3, LSM-780, Zeiss) and the fluorescent images were captured in a single Z-slice for figure construction.

## Dot blot assay

Approximately $1 \times 10^7$ trophozoites were vigorously vortexed in 200 µl of lysis buffer (1% Triton X-100, 100 µg/ml TLCK, 1× Protease inhibitor cocktail, 1× phosphatase inhibitor cocktail, 5 mM EDTA in TBS) at 4˚C for 20 min to prepare the total protein lysate. The serial-diluted protein lysates were blotted on the nitrocellulose (NC) membrane (ThermoFisher Scientific) by a 96-well microfiltration apparatus (BIO-RAD), then the NC membrane was blocked in blocking buffer (5% nonfat milk in TBS with 0.1% Tween-20) with gentle agitation at 37˚C for 1 hr. After triple washes with TBS containing 0.1% Tween-20 (TBST), the membrane was incubated with the primary mouse anti-PIP$_2$ (4000×, Echelon Biosciences) or mouse anti-α-tubulin (Sigma Aldrich, DM-1A) antibodies at 4˚C overnight. Then, the membrane was washed with TBST three times, followed by incubation with HRP-conjugated goat anti-mouse IgM or IgG secondary antibodies at 37˚C for 1 hr. The signal was detected by an enhanced chemiluminescence substrate (ThermoFisher Scientific) and imaged by a UVP imaging system (Analytik Jena Company).

## Western blotting

The protein sample (30µg/lane) denatured in 1× SDS sample buffer was separated by sodium dodecyl sulfate polyacrylamide gel electrophoresis (SDS-PAGE) in a 12% gel and blotted onto the polyvinylidene difluoride (PVDF) membrane by a tank transfer system (BIO-RAD). The membrane was incubated in the blocking buffer (5% nonfat milk in TBST) at 37˚C for 1 hr and then incubated with primary antibodies, mouse anti-HA (5000×, Sigma-Aldrich), rabbit anti-*Tv*PI4P5K (15000×), mouse anti-α-actin (10000×, Genetex), mouse anti-GAPDH (5000×), mouse anti-α-tubulin (10000×, Sigma-Aldrich), and rat anti-*Tv*CyP2 (3000×) [36] diluted in blocking buffer, at 4˚C overnight. After optimal washing by TBST, the membrane was reacted with HRP-conjugated goat anti-mouse, rat, or rabbit IgG secondary antibody (5000×, Jackson ImmunoResearch) prepared in the blocking buffer, at 37˚C for 1 hr. The signal was detected by LimiFlash Ultima Chemiluminescent Substrate (VISUAL PROTEIN). To detect endogenous *Tv*PI4P5K expression, 80 µg of protein lysate and a high-sensitive Femto Chemiluminescent substrate (Visual Protein) were used for western blotting detection.

## *In vitro* phospholipid kinase assay for *Tv*PI4P5K

The *in vitro* PI4P5K activity assay kit (Echelon Biosciences, K-5700) is an ATP depletion assay that quantifies the remaining ATP in the sample following the kinase reaction. The PIP5K activity was measured according to the manufacturer's instructions (Echelon Biosciences). Briefly, 10 µl of the different concentrations of recombinant His-*Tv*PI4P5K (1–414 amino acid) wild type or K136A protein in 1× KBZ buffer (Echelon Biosciences) was mixed with 10 µl of 4× PI(4)P substrate solution [400 µM PI(4)P] (Echelon Biosciences) before the addition of 20 µl of 2× ATP solution (20 µM ATP) (Echelon Biosciences). The sample was reacted at 37˚C for 2 hrs, then 40 µl of ATP detector was added to each well for 20 min at room temperature in the dark, and the luminescent signal was detected by a spectrophotometer at a filter wavelength of 550 nm [35,64].

## *In vivo* G-actin/ F-actin fractionation

A commercial *in vivo* assay biochem kit (Cytoskeleton Inc) was used to fractionate G- and F-actin, according to the operating manual, with minor modifications. Briefly, approximately 3×

$10^7$ trophozoites were incubated in cell lysis buffer (Cytoskeleton Inc) with vigorous agitation at 4°C for 30 min and homogenized using a 23-gauge needle on a 5-ml syringe. Next, the total lysate was centrifuged at 1000× *g* to remove the cell debris, followed by ultracentrifugation at 100000× *g* for 1 hr to recover the insoluble F-actin in the pellet from soluble G-actin in the supernatant. In western blotting, α-tubulin and *Tv*CyP2 were detected as purity markers for the supernatant and pellet fractions, respectively. The α-actin signal intensity in the supernatant and pellet fractions were first normalized to *Tv*CyP2 and α-tubulin, respectively, and then the α-actin signal ratio between supernatant and pellet fractions was calculated to evaluate actin assembly [28].

## Morphogenesis analysis

The flagellate trophozoites were cultured in a T25 flask at 37°C for 30 min, and the parasite morphology was observed by microscopy at phase-contrast mode (Olympus, CKX31). The flagellate trophozoites had a solid spherical shape and diameter under 10 μm, whereas the amoeboid trophozoites had a diameter over 10 μm and a distinctive irregular appearance or flat round disk form laying on the glass surface. The percentage of flagellate or amoeboid forms was measured in 300 trophozoites within five random microscopic fields. Dynamics of parasite amoeboid morphogenesis were recorded by time-lapse microscopy with a capturing rate of 30 sec per frame.

## Cytoadherence assay

The trophozoites were incubated with 5 μM CFSE (ThermoFisher Scientific) in PBS with 1% BSA on a gentle rotation at 37°C for 30 min. After washing three times with PBS to remove the excess dye, the parasites were suspended in keratinocyte serum-free medium (Gibco) for analysis. The CFSE-labeled parasites were co-cultured with *h*VECs at the multiplicity of infection (MOI) of 2 in a minimal thin layer of keratinocyte serum-free medium for 1 hr in 5% CO$_2$. After washing the unbound trophozoites away with pre-warmed PBS, the CFSE signal was detected by inverted fluorescence microscopy (CFSE Ex/Em = 492/517) (Axiovert 200M, Zeiss). For each assay, the signal intensities quantified by ImageJ v.1.53k software (National Institutes of Health) in five independent microscopic fields were averaged to evaluate the relative parasite cytoadherence. The signal from the control parasites was defined at 100%.

## Cytolysis by fluorescence microscopy

The trophozoites were incubated with 5 μM Orange-CMRA (ThermoFisher Scientific) in PBS with 1% BSA on a gentle rotation at 37°C for 30 min. After washing three times with PBS to remove the excess dye, the parasites were suspended in keratinocyte serum-free medium (Gibco) for analysis. The Orange-CMRA-labeled TH17 trophozoites and *h*VECs were co-cultured at MOI of 4 on a coverslip placed in a culture microplate with keratinocyte serum-free medium. The microplate was centrifuged at 200× *g* for 5 min to sediment trophozoites for contact with the *h*VECs monolayer. After removing unbound trophozoites by three washes with medium, samples were incubated in keratinocyte serum-free medium under the atmosphere with 5% CO$_2$ at 37°C for 1 hr. The specimens fixed with 4% formaldehyde, then permeabilized with PBS containing 0.2% Triton X-100, were stained with FITC-conjugated Phalloidin (1000×, Abcam, ab235137) at room temperature for 1 hr. The samples were washed three times with PBS and air-dried for 20 min. The coverslips were mounted in an anti-fade medium with DAPI and inverted on a glass slide for observation by confocal microscopy (FITC Ex/Em = 492/518, Orange-CMRA Ex/Em = 548/576) (LSM700, Zeiss). The clear lytic area was

measured by AxioVision software (Rel.4.8, Zeiss). For each assay, the lysis area per trophozoite was averaged from five independent microscopic fields to evaluate parasite cytolysis activity.

## Lactate dehydrogenase (LDH) cytotoxicity assay

The host cells incubated in a 96-well microplate at 90% confluence were inoculated with the parasite in keratinocyte serum-free medium in 5% $CO_2$ for 1 or 4 hrs. The spent media collected at different time points post-infection were centrifuged at 1000× *g* to remove the cells or parasites before analysis. The supernatant was assayed using the LDH cytotoxicity assay kit (Biochain). Briefly, 45 μl of Assay Mixture was added to 100 μl of test sample for 30 min at room temperature, then 50 μl of Stop Solution was added, and the colorimetric signal was detected at OD$_{490}$ on a spectrophotometer (SpectraMax190, Molecular Devices). For each assay, the samples collected from parasite-free host cells treated with or without Lysis Solution were detected as high- and low-level controls, respectively. The cytotoxicity (%) was measured as follows: $\frac{OD490\ (test\ sample) - OD490\ (low-level\ control)}{OD490\ (high-level\ control) - OD490\ (low-level\ control)} \times 100$.

## IP$_3$ detection

Approximately $3\times 10^7$ trophozoites suspended or scratched in 500 μl of 4°C PBS were lysed by ultrasonication on ice. The cell lysates were pelleted by 15000× *g* centrifugation to remove cell debris and insoluble particles, then diluted in Sample Diluent (Abcam) for colorimetric quantification with an IP$_3$ ELISA kit (Abcam) at OD$_{450}$.

## Subcellular fractionation by differential and gradient centrifugation

Organelle fractions were purified from 250 ml of cells for biochemical characterizations by differential and gradient centrifugation as previously described with some modifications [36]. Briefly, the postnuclear lysate was processed by differential centrifugation into crude membrane fractions, P15 and P100, and the soluble S100 fraction. The P100 pellet was re-suspended in 0.5 ml of TBS by sonication, mixed with 0.1 ml of 60% OptiPrep, and layered onto an OptiPrep gradient gel (12~30%), which was formed by a step-wise 2% increase in each layer. Samples were centrifuged at $3.53\times10^5$ *g* at 4°C for 4 hr (Beckman SW60 rotor). The sample was fractionated into 200-μl fractions starting from the top of the gradient.

## Statistical analysis

The data were analyzed using Microsoft Office Excel 2019 software with Student's t-test. A p< 0.05 was considered as a significant difference.

## Supporting information

**S1 Fig. Cytoadherence of the clinical isolates.** The CFSE-labeled trophozoites from G3 and NTU252, NTU258, and NTU285 clinical isolates, were co-cultured with *h*VECs for cytoadherence assay. The CFSE signal was recorded by fluorescence microscopy and quantified by calculating the number of adherent parasites to 1000 *h*VECs. The assay was processed in three biological repeats (n = 3, mean ± SD). The significant differences for the paired samples were analyzed by Student's t-tests, with *p*< 0.05(*), *p*< 0.01(**), ns. no significant difference. (TIF)

**S2 Fig. Protein sequence alignment for *Tv*PI4P5K.** The alignment of human *h*PI4P5K (Q99755), *T. vaginalis* *Tv*PI4P5K (TVAG_462290), and *Tv*PI4P5K-2 (TVAG_456620), with identical or similar amino acids highlighted. The putative sites for ATP interaction (*),

membrane binding (+), PI4P substrate binding (†), Mn$^{2+}$ and Mg$^{2+}$ binding (●), and the kinase-activity-essential K136 residue (▾) are indicated at the top of sequences.
(TIF)

**S3 Fig. The partial colocalization of PIP$_2$ and *Tv*PI4P5K at the plasma membrane.** The IFA images from Fig 2E and 2I were viewed by 2.5D view of ZEN software as shown in (A) and (B), respectively. The intensities in the two-dimensional image were converted into a height map and are represented by the extension in the Z-direction. RF indicates recurrent flagellum and PM indicates plasma membrane. The yellow arrowheads indicate colocalization (yellow) of *Tv*PI4P5K (red) and PIP$_2$ (green) in particular plasma membrane regions. (C) P100 was fractionated by OptiPrep density gradient ultracentrifugation (left panel) and 200-μl aliquots were collected from the top of each gradient for western blotting using antibodies as indicated. Myb1, Myb3IP$_{hmw}$, and *Tv*Gα protein were detected as membrane compartment markers for P100/Myb1, P100/Myb3IP$_{hmw}$, and the plasma membrane (PM), respectively.
(TIF)

**S4 Fig. The partial plasma membrane colocalization of PIP$_2$ and *Tv*PI4P5K.** The IFA images from Fig 3D and 3E were viewed by 2.5D view of ZEN software as shown in (A) and (B), respectively. The intensities in a two-dimensional image were converted into a height map and represented by the extension in the Z-direction. The yellow arrowheads indicate colocalization (yellow) of *Tv*PI4P5K (red) and PIP$_2$ (green) at specific plasma membrane regions.
(TIF)

**S5 Fig. IP$_3$ production upon *T. vaginalis* morphogenesis.** The cell lysates from trophozoites before (0′) and after amoeboid transition (30′) were detected with a commercial IP$_3$ ELISA kit, and the relative colorimetric signal was analyzed by a spectrophotometer at OD$_{450}$. The assay was processed in three biological repeats (n = 3, mean ± SD). The significant differences for the paired samples were analyzed by Student's t-tests, with $p < 0.05$(*), $p < 0.01$(**), ns. no significant difference.
(TIF)

**S6 Fig. Edelfosine inhibits PIP$_2$ reduction in the parasites with amoeboid morphogenesis.** The protein lysates extracted from the non-transgenic and transgenic trophozoites with or without Edelfosine treatment before (0′) and after culture in a T25 flask for 30 min (30′) were subjected to dot blot for PIP$_2$ and α-tubulin detection. The assay was processed in three biological repeats, and the relative PIP$_2$ signal intensity normalized to α-tubulin is shown in the bar graph (n = 3, mean ± SD). The significant differences for the paired conditional samples were analyzed by Student's t-tests, with $p < 0.05$(*), $p < 0.01$(**), ns. no significant difference.
(TIF)

**S7 Fig. Drug effects on *T. vaginalis* viability.** The viability of *T. vaginalis* trophozoites treated with DMSO, Edelfosine, BAPTA-AM, or EGTA was analyzed using the Trypan blue exclusion assay. The assay was processed in three biological repeats (n = 3, mean ± SD). The significant differences for the paired samples were analyzed by Student's t-tests, with $p < 0.05$(*), $p < 0.01$(**), ns. no significant difference.
(TIF)

**S8 Fig. Extracellular calcium involves PIP$_2$-dependent *T. vaginalis* morphogenesis.** The CG-preloaded TH17 flagellates were inoculated in the medium with or without EGTA and incubated on a glass slide for 30 min. The parasites before (0′) and after (30′) morphogenesis were fixed for CG detection (A) or morphogenesis assay (B). (A) The relative CG signal intensity was quantified as shown in the bar graph. (B) The percentage of flagellate versus amoeboid

trophozoites was measured as shown in the bar graph. Black and white arrowheads mark amoeboid and flagellate trophozoites, respectively. The assays were processed in three biological repeats (n = 3, mean ± SD). Significant differences were statistically measured by Student's t-tests, with $p < 0.05(*)$, $p < 0.01(**)$, and ns, no significant difference.
(TIF)

**S9 Fig. A23187 induces *T. vaginalis* intracellular calcium and actin polymerization.** (A) The TH17 trophozoites treated with DMSO or A23187 were loaded with CG for confocal microscopy. The relative intensity of the CG signal was quantified in 300 trophozoites from five independent microscopic fields as shown in the bar graph (n = 3, mean ± SD). (B) The TH17 trophozoites pretreated with DMSO or A23187 were fractionated into supernatant and pellet fractions for western blotting. The ratio of α-actin signals in the supernatant (S) versus pellet (P) fractions was quantified, as shown in the bar graphs. The assays were processed in three biological repeats (n = 3, mean ± SD). Significant differences were statistically measured by Student's t-tests with $p < 0.05(*)$, $p < 0.01(**)$, and ns, no significant difference.
(TIF)

**S10 Fig. Iron triggered *Tv*PI4P5K plasma membrane localization.** (A) The TH17 trophozoites cultured in normal-iron, iron-depleted, and iron-repletion medium overnight were fixed for IFA double staining for *Tv*PI4P5K (red) and PIP$_2$ (green) detection. The nuclei were stained with DAPI. The images boxed are magnified as shown in plots (a-c). Scale bar: 2 μm. White arrowheads indicate the parasite plasma membrane border (PM), and C indicates the cytoplasm. (B) The IFA images from (A) were viewed by 2.5D view of ZEN software. The yellow arrowheads indicate the partially colocalized signals (yellow) of PIP$_2$ (green) and *Tv*PI4P5K (red) in specific plasma membrane areas. (C) Transcription of the *tvpi4p5k* gene in *T. vaginalis* from various iron conditions was quantified by qPCR. The relative gene expression normalized to *β-tubulin* was shown in the bar graph (n = 3, mean ± SD). Significant differences were analyzed by Student's t-tests, with $p < 0.05(*)$ and $p < 0.01 (**)$.
(TIF)

**S11 Fig. A23187 does not affect *Tv*PI4P5K-dependent PIP$_2$ production.** The iron-depleted TH17 trophozoites with or without the A23187 challenge were fixed for IFA double staining with anti-*Tv*PI4P5K and anti-PIP$_2$ antibodies. Nuclei were stained with DAPI. Scale bar: 2 μm. The signal intensity distribution on the yellow line between x and y sites in the representative micrograph (a, b) was analyzed by ImageJ as shown in the corresponding plots (a, b). PM indicates the plasma membrane boundary.
(TIF)

**S12 Fig. Culture conditions for parasites or *h*VECs viabilities.** (A) The viability of TH17 trophozoites in a microplate with keratinocyte serum-free medium at 37°C in 5% CO$_2$ was evaluated at different time points by the trypan blue exclusion assay. (B) The *h*VECs were co-cultured with the parasites with conditions as in Fig 8A at different MOI for 4 hr. The spent medium supernatant was collected for LDH cytotoxicity assay. (C) The parasites were pre-treated as in Fig 8A. Different amounts of trophozoites (equivalent to the trophozoites number used in Fig 8A) were inoculated in keratinocyte serum-free medium for 1 hr at 37°C under 5% CO$_2$. The spent medium supernatant was collected for LDH cytotoxicity assay. (D) The spent media from the parasite-free *h*VECs culture treated with DMSO or PIC for 1 or 4 hr were collected for LDH cytotoxicity assay. The assays were processed in three biological repeats (n = 3, mean ± SD), and significant differences were analyzed by Student's t-tests, with $p < 0.05(*)$ and $p < 0.01 (**)$.
(TIF)

**S13 Fig. Effects of HA-*Tv*PI4P5K overexpression in the less adherent isolate.** T1 trophozoites with or without HA-*Tv*PI4P5K overexpression were sampled for IFA double staining with anti-HA or anti-PIP$_2$ (A), morphogenesis assay (B), and cytoadherence assay (C). For (B), white arrowheads label flagellate trophozoites. The assay was processed in three biological repeats to measure the proportion of the flagellate versus amoeboid trophozoites, as shown in the bar graph (n = 3, mean ± SD). For (C), the CFSE-prelabeled parasites were co-cultured with *h*VECs for 1 hr. After washing, the bound parasites were detected by a confocal microscope. The assay was processed in three biological repeats, and the relative ratio of the bound parasite was quantified with the non-transgenic control defined as 1 (n = 3, mean ± SD). The significant differences were analyzed by Student's t-tests, with $p < 0.05(^*)$, $p < 0.01(^{**})$, ns. no significant difference.
(TIF)

**S1 Movie. Dynamics of *T. vaginalis* amoeboid morphogenesis.** The TH17 flagellate trophozoites were cultured in a glass slide at 37˚C. Dynamics of flagellate-amoeboid transition was recorded by time-lapse microscopy with a capturing rate of 30 sec per one frame for 30 min.
(MP4)

**S2 Movie. Dynamics of CG signal in the parasite with amoeboid transformation.** The DMSO- or Edelfosine-pretreated TH17 flagellate trophozoites were loaded with CG and inoculated in the medium with or without EGTA. The conditional samples were cultured on a glass slide at 37˚C. Dynamics of CG signal intensity in the transforming trophozoites were recorded by time-lapse fluorescence microscopy with a capturing rate of 40 sec per one frame for 20 min. Scale bar: 10 μm.
(MP4)

**S1 Data. Raw Blots.** This file includes raw image data of Figs 1C, 2A, 2C, 2G, 2H, 3F, 5A, 5B, 5C, 5D, 5E, 6A, 6B, 7D, S3C, S6 and S9B.
(PDF)

**S2 Data. Statistical Analysis.** This file includes raw data of statistical analysis in Figs 1C, 1D, 2B, 2D, 2E, 2H, 2I, 3B, 3C, 3D, 3E, 3F, 4A, 4B, 4C, 4D, 4E, 5B, 5C, 5D, 5E, 6A, 6B, 6C, 7A, 7B, 7C, 7D, 8A, 8B, S1, S5, S6, S7, S8A, S9A, S9B, S10C, S11, S12A, S12B, S12C, S12D, S13B and S13C.
(XLSX)

## Acknowledgments

We are grateful to Dr. Jung-Hsiang Tai (Institute of Biomedical Sciences, Academia Sinica, Taiwan) for providing the *T. vaginalis* T1 isolate. We thank the imaging core at the First Core Labs, National Taiwan University College of Medicine, for the technical support in image acquisition and analysis.

## Author Contributions

**Conceptualization:** Hong-Ming Hsu.

**Funding acquisition:** Hong-Ming Hsu.

**Investigation:** Yen-Ju Chen, Kuan-Yi Wu, Shu-Fan Lin, Hong-Ming Hsu.

**Methodology:** Yen-Ju Chen, Kuan-Yi Wu, Shu-Fan Lin, Sung-Hsi Huang, Heng-Cheng Hsu.

**Project administration:** Hong-Ming Hsu.

**Supervision:** Hong-Ming Hsu.

**Validation:** Yen-Ju Chen, Kuan-Yi Wu, Shu-Fan Lin, Sung-Hsi Huang, Heng-Cheng Hsu, Hong-Ming Hsu.

**Writing – original draft:** Hong-Ming Hsu.

**Writing – review & editing:** Hong-Ming Hsu.

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
