## [Decision Letter · Decision Letter 0]

13 Sep 2023

Dear Dr Hsu,

Thank you for submitting your manuscript titled “PIP2 regulating calcium signal modulates actin cytoskeleton-dependent cytoadherence and cytolytic capacity in the protozoan parasite Trichomonas vaginalis” to PLoS Pathogens for consideration.

The manuscript was reviewed by members of the editorial board and three expert reviewers, whose full reviews are included at the end of this email.

In general, all reviewers welcome the data shown in this submission and consider it worthy of dissemination to the wider scientific community. However, two of the three reviewers list significant concerns with respect to some of the presented experimental approaches, the data derived from these and data interpretation.

Reviewer 1 questions the appropriateness of approaches to quantification of expression and makes concrete suggestions for improvement. Reviewer 3 is especially concerned with approaches to the monitoring of cytotoxicity and with experimental conditions for cytoadherence studies, a concern also shared by Reviewer 1.

For these and other reasons related to clarity of text, usage of acronyms and figure labelling, the manuscript can be considered for publication only after major revision.

We cannot make any decision about publication until we have seen the revised manuscript and your response to the reviewers' comments. Your revised manuscript is also likely to be sent to reviewers for further evaluation.

Sincerely,

Carmen Faso, Ph.D.

Academic Editor

PLOS Pathogens

James Collins III

Section Editor

PLOS Pathogens

Kasturi Haldar

Editor-in-Chief

PLOS Pathogens

orcid.org/0000-0001-5065-158X

Michael Malim

Editor-in-Chief

PLOS Pathogens

orcid.org/0000-0002-7699-2064

**Reviewer's Responses to Questions**

**Part I - Summary**

Reviewer #1: TV is the most common cellular pathogen sexually transmitted worldwide and it is particularly common among resource limited populations. TV infections are strongly associated with higher transmission rate of HIV and a number of other important health sequelae including low birth weight, premature rupture of membranes and preterm. However, we still know very little about the pathogenesis of this common parasite. Here, the authors demonstrate that PIP2 hydrolysis followed by intracellular calcium rising is vital for parasite amoeboid morphogenesis and cytoadherence. I consider that there is valuable information on the paper that will likely be of interest to the T. vaginalis community because there is still very little known about the molecular basis of host interaction for this parasite. However, the some of the data presented and its interpretation require modifications prior to publication. The resubmission of the manuscript is encouraged after major modification.

Reviewer #2: This work satisfies the PLOS Pathogens criteria for publication: The manuscript by Chen et al., entitled “PIP2 regulating calcium signal modulates actin cytoskeleton-dependent cytoadherence and cytolytic capacity in the protozoan parasite Trichomonas vaginalis” is an interesting piece of data that includes correctly design experiments, clean result presentation, and great novelty for the field of host-parasite interaction. In this work, the authors investigated in deep the link between the cytoskeleton and PIP2 signaling in T. vaginalis and the possible relationship with environmental iron response. This study identified differential PIP2 and TvPI4P5K expression in adherent and nonadherent Tv isolates and analyzed their functional roles in the cytopathogenicity of Tv. Their data reveal the biological significance of PIP2 signal transduction in regulating T. vaginalis pathogenicity.

Reviewer #3: The authors investigated PIP2 distribution in a human parasite Trichomonas vaginalis, its role as a second messenger, the role of phosphatidylinositol-4-phosphate 5-kinase (TvPI4P5K), calcium and iron in flagellate-amoeboid morphogenesis, and relationship to the cell virulence (adherence, cytotoxicity). They showed that PIP2 localizes in the undulating membrane of non-adhering cells (T1), while on the plasma membrane of adhering cells (TH17). They confirmed the expected activity of TvPI4P5K and claimed that it colocalized on the plasma membrane with PIP2 in adherent cells in which higher expression was observed, while TvPI4P5K was in the cytosol of nonadhering cells (T1). They further reported that expression of mutated HA-tagged TvPI4P5K has a dominant negative effect on trichomonad's ability to transform to amoeboid form, adherence to hVEC cells, intracellular calcium level, and polymerization of actin. Moreover, iron depletion conditions were shown to affect the localization of PIP2, TvPI4P5K, and calcium. Finally, the authors claimed that expression of HA-tagged TvPI4P5K stimulated cytotoxicity, while it was reduced in K136A mutant. The study is of moderate technical quality, the experiments are well-controlled and easy to follow, however, some conclusions are not supported by the data.

**Part II – Major Issues: Key Experiments Required for Acceptance**

Reviewer #1: Specific comments

1. In order to make it easier to follow the rational behind the experiments, it would be nice to include schematic illustration of the complete PIP2 pathway. Without this illustration the results are difficult to follow.

2. Fig 1E: Quantification needs to be done using a more quantitative method such as quantitative PCR (qPCR) or similar.

3. Page 13, line 171: ¨HA-TvPI4P5K Wt and K136A were five-fold overexpressed in TH17 trophozoites (Fig. 2G)¨. Fig 2G is an immunoprecipitation experiment. It is not the appropriate experiment to measure expression. As example, it could be the case that endogenous protein could be forming complex but not the HA-tag version, then, you can pull down more HA-TvPI4P5K due to an accessibility issue.

4. Fig 2G: the image from the anti-Tv PI4P5K antibody was cut and the signal cannot be visualized properly. Please, modify and show the complete image to allow a better interpretation.

5. Page 16, line 217: ¨~80% trophozoites of TH17 non-transfectant or HA-TvPI4P5K transfectant transformed into amoeboid form but only ~40% in K136A mutant¨. Similar results were obtained when adherence was evaluated. I understand that no difference in amoebic transformation and adherence is observed when HA-TvPI4P5K is overexpressed probably because TH17 is already an adherent strain. To validate the obtained results, it would be nice to evaluate if there is a difference in amoebic transformation and parasite adherence when HA-TvPI4P5K is overexpressed in G3 or NTU285 less adherent isolates

6. In order to confirm that amoebic transformation induce PIP2 hydrolysis, it would be important to evaluate the level of PIP2 (similar as Fig 3F) using parasites transfected with HA-TvPI4P5K and K136A compared to control (experiment Fig 3B)

7. Fig 5A: Why endogenous TvPI4P5K is not detected in the IP when using transfected parasites? (Although the figure was cut, it seems that endogenous protein was detected in Fig 2G)

8. Page 26, line 333: ¨The dot blot (Fig. 6A) and western blotting (Fig. 6B) revealed that the PIP2 and TvPI4P5K levels similar in the normal-iron or iron-replete parasites were higher than the parasites depleted of iron, showing that iron content in the normal growth medium sufficiently affects TvPI4P5K and PIP2 expression in T. vaginalis¨. This is a similar comment as before. I think that an IP is not the correct experiment to evaluate expression. Although a lower expression might be a possibility, there might be also alternative explanations: iron might be affecting the complex formation that might be altering the accessibility of the protein to the antibody affecting the results of the IP. The expression of TvPI4P5K should be demonstrated in the total extract, the IP might be affected by other circumstances. Based on these results, I don´t agree with this statement.

9. Page 26, line 342: ¨ suggesting that iron very likely modulates TvPI4P5K expression…¨. To demonstrate that TvPI4P5K gene expression is affected in iron-depleted, and iron-replete growth media, a qPCR analysis must be included. It is an easy experiment that can help to demonstrate if expression is affected by iron.

10. Fig 8: What does ¨The conditional parasites¨ means? was this experiment performed with parasites or conditioned media? I am not sure if I am understanding the experiment correctly, I assume that the experiment was performed using parasites.

11. I am surprised to see that cytolysis experiments were performed in 1 h incubation. Usually, to evaluate cytolysis, the parasites are incubated with host cells for longer times (minimum 4 h in Lustig et al 2013, 8 h in Bastida Corcuera et al. 2005). The level of cytolysis (even when using highly cytolytic strains) is usually very low after 1 h incubation. What is the level of cytolysis observed in these experiments? Fig 8A is normalized and does not allow to evaluate the level of cytolysis. It might be worthy to perform this experiment using longer incubation time to increase cytolysis. Also, as control, it is important to evaluate the cytolysis of host cells when incubated with the correspondent media plus the different drugs (without parasites). It is important to include this control as these drugs might have a direct effect in the host cells

Finally, I recognize that the authors are not native English speakers, and I hesitate to say so because I appreciate the authors' efforts to produce an article in a language other than their own, but the material is difficult to read and the grammar needs to be improved. The authors conducted a thorough work, and enhancing the English will make the paper much more enjoyable to read.

Reviewer #2: No new experiments or modifications are required.

Reviewer #3: The cell localization of TvPI4P5K is not convincing. Fig. 2E, TvPI4P5K in TH17 is possibly on the membrane, however, why did the authors select just the displayed cell with a very strong signal only on one site? How many cells displayed this pattern? In Fig. 3D, TvPI4P5K is mainly in the cytosol. PCC supported some colocalization with PIP2, however, this is not because of colocalization on the membrane, but also colocalization in the cytosol. Fig 3E. The effect of Edelfosine, the membrane localization of TvPI4P5K is not convincing. The yellow line is drawn in the area with membrane localization but there is a significant cytosolic signal in the other part of the cell. In general, for colocalization, it is preferable to use 2D histogram for channels to cover the whole cell rather than a line (yellow lines in Figures) as the result depends on the line position. The authors should provide more convincing IF figures for TvPI4P5K and if possible westerns of cell fractions.

396. The authors used LDH assay to monitor the cytotoxicity of trichomonads on hVECs. However, this test is not suitable for T. vaginalis as there is a strong LDH activity in the parasite cytosol. It is impossible to distinguish the origin of LDH activity upon coincubation of T. vaginalis with hVECs. Moreover, if the atmosphere contains oxygen during the coincubation, the dying of trichomonad is significant. This experiment should be performed using more specific detection such as radiolabelling of target cells.

219. The conditions of the cytoadherence test are not provided in sufficient detail to evaluate its relevance. Which medium was used for coincubation and under which atmosphere it was performed? This is critical information as trichomonads are sensitive to oxygen and incubation under aerobic conditions will induce oxygen-dependent changes in trichomonad physiology and apoptosis which compromises these results.

412 and 698. Cytoadherence study. Similarly as above. In which atmosphere trichomonads and hVEC cells were incubated in a minimal thin layer of culture media for 1 hr? If the atmosphere was aerobic, they were dying during the incubation and the experiments provide no relevant information.

**Part III – Minor Issues: Editorial and Data Presentation Modifications**

Reviewer #1: Minor comments:

- Page 13, line 167: ¨The data and was consistent with the earlier RT-PCR (Fig. 1F), supporting the potential correlation of TvPI4P5K and PIP2 expression with T. vaginalis cytoadherence¨. It should be Fig 1E

- Replace ¨nonadherent isolates¨ by ¨less adherent isolates.

- Page 20, line 271. Introduce that BAPTA-AM is a cell-permeant chelator that can be used to control the level of intracellular Ca 2+.

- Explain what CG is in the figure legend 4

- Page 24, line 312: explain that ¨A23187¨ is a Calcium Ionophore

Reviewer #2: There are few minor issues that need to be considered to improve clarity, such as:

1. In Fig. 1A please explain what are describing the 2nd part of the figure in which the signal intensisty is so weak. Are they representing negative controls? Please include the missing information in the corresponding figure legend.

2. P7, line 167. Please chech this sentence. It appears incomplete.

3. P7, line 184. …His-TvPI5P5K… It should be His-TvPI4P5K? Please correct it as need it.

4. P17, lines 267, 269. …GC… It should be CG (Calcium green). Please correct them as needed.

5. Line 487 and S6 line 66. …TvPI5P5K…It should be TvPI4P5K?

Reviewer #3: Fig 2. Why did authors coIP HA-tagged TvPI4PK? Is the expression from the vector low that the protein cannot be detected directly on the western blot of cell lysate or cell fractions? Based on HA tag signal in IFA, it should be detectable on westerns.

Fig 7. The authors showed that A23187 reversed several effects of iron depletion, including calcium content, transformation to amoeboid forms, cytotoxicity etc. This is a very interesting result, but it should be discussed in more detail with available publications reporting the effect of iron on trichomonad gene expression and physiology. The discussion line 514 and following “Iron and signal transduction“ is very limited.

line 219. The author should specify in the text that trichomonads were coincubated with hVEC cells to test the cytoadherence.

Fig 1. D. The alignment is more suitable for supplementary materials.

Material and methods.

line 545. The authors should specify how the T. vaginalis strains were cryopreserved and to which publicly available culture collection were deposited.

563. The plasmid must be clearly described. Reference 60 (line 568) provides no information but another reference for the article in which the plasmid is not described in detail either. It is important to provide information on the FLP promotor, how strong the gene expression is upon this promotor etc.

624. The authors used a confocal microscope, however, there is no information on how many sections were performed, their size, and how many sections were used for the figure construction.

698. The CFSE labeling needs a description.

706. Cytolysis. I miss the information about the atmosphere and medium which was used for co-incubation.

Fig E, I. The “merge“ of IF with DIC is superfluous. The same for double merge figures with and without lines indicating sections for signal intensity distribution. A single merge with lines would be sufficient, but 2D histogram instead of lines would be more informative.

271 Please specify for readers in the text that BAPTA-AM is a calcium chelator.

369. Please specify in the text what is A23187 - calcium ionophore

196. Please, spell out FLP.

400. Please, sell out MOI.

The authors should be more careful with the interpretation of the effect of K136A mutation as an effect of PIP2 signaling: subtitle 303, 329, 396, and the title.

PLOS authors have the option to publish the peer review history of their article (what does this mean?). If published, this will include your full peer review and any attached files.

Reviewer #1: No

Reviewer #2: No

Reviewer #3: No
---

## [Decision Letter · Decision Letter 1]

8 Dec 2023

Dear Dr Hsu,

We are pleased to inform you that your revised manuscript 'PIP2 regulating calcium signal modulates actin cytoskeleton-dependent cytoadherence and cytolytic capacity in the protozoan parasite Trichomonas vaginalis' has been provisionally accepted for publication in PLOS Pathogens.

Best regards,

Carmen Faso, Ph.D.

Academic Editor

PLOS Pathogens

James Collins III

Section Editor

PLOS Pathogens

Kasturi Haldar

Editor-in-Chief

PLOS Pathogens

orcid.org/0000-0001-5065-158X

Michael Malim

Editor-in-Chief

PLOS Pathogens

orcid.org/0000-0002-7699-2064

Reviewer Comments (if any, and for reference):

Reviewer's Responses to Questions

**Part I - Summary**

Reviewer #1: (No Response)

Reviewer #2: This work shows novelty and significance in the field of T. vaginalis pathogenicity. A new dimension in the study of T. vaginalis pathogenicity has been reach. For example, specific TvPI4P5K and PIP2 expression was observed in the T. vaginalis adherent isolate. During parasite flagellate-amoeboid transition, cell membrane PIP2 was cleaved by a PLC dependent pathway, increasing intracellular calcium essential for cytoskeleton activities, including actin remodeling, morphogenesis, and cytoadherence, which could be inhibited by Edelfosine or BAPTA and activated by A23187. When the iron content in the normal growth medium sufficiently elicited PIP2 signaling, iron simultaneously regulated the expression and plasma membrane localization of TvPI4P5K for PIP2 production. In the host-parasite interaction, PIP2 signaling cascades modulated parasite morphogenesis and cytoadherence, contributing to the cytopathic effects by extracellular protease-associated cytolysis in an actin cytoskeleton dependent manner. The PIP2-triggered extracellular cytopathic effectors warrant continued identification but this study provides an insight into a new dimension of T. vaginalis pathogenicity.

The modified manuscript has been improved and the authors have cosidered all the reviewers commments and suggestions. Thus, I consider that it is now ready for publication.

**Part II – Major Issues: Key Experiments Required for Acceptance**

Reviewer #1: (No Response)

Reviewer #2: No new experiments are required.

**Part III – Minor Issues: Editorial and Data Presentation Modifications**

Reviewer #1: (No Response)

Reviewer #2: No minor issues or modifications are required in the modified manuscript.

PLOS authors have the option to publish the peer review history of their article (what does this mean?). If published, this will include your full peer review and any attached files.

Reviewer #1: No

Reviewer #2: No

---

## [Editor Report · Acceptance letter]

11 Dec 2023

Dear Dr Hsu,

We are delighted to inform you that your manuscript, " PIP2 regulating calcium signal modulates actin cytoskeleton-dependent cytoadherence and cytolytic capacity in the protozoan parasite Trichomonas vaginalis ," has been formally accepted for publication in PLOS Pathogens.

Best regards,

Michael Malim

Editor-in-Chief

PLOS Pathogens

orcid.org/0000-0002-7699-2064